# KinDEL: DNA-Encoded Library Dataset for Kinase Inhibitors

## Abstract

DNA-Encoded Libraries (DEL) are combinatorial small molecule libraries that offer an efficient way to characterize diverse chemical spaces. Selection experiments using DELs are pivotal to drug discovery efforts, enabling high-throughput hit finding screens. However, limited availability of public DEL datasets hinders the advancement of computational techniques designed to utilize such data. To bridge this gap, we present **KinDEL**, one of the first large, publicly available DEL datasets on two kinases: Mitogen-Activated Protein Kinase 14 (MAPK14) and Discoidin Domain Receptor Tyrosine Kinase 1 (DDR1). Interest in this data modality is growing due to its ability to generate extensive supervised chemical data that densely samples around select molecular structures. Demonstrating one such application of the data, we benchmark different machine learning techniques to develop predictive models for hit identification; in particular, we highlight recent structure-based probabilistic approaches. Finally, we provide biophysical assay data, both on- and off-DNA, to validate our models on a smaller subset of molecules. Data and code for our benchmarks can be found at `https://kin-del-2024.s3.us-west-2.amazonaws.com/kindel.zip`.

## 1 Introduction

DNA-Encoded Libraries (DEL) have emerged as a powerful tool in drug discovery, enabling highly efficient screens of small molecule libraries against therapeutically relevant targets (Yuen & Franzini, 2017; Gironda-Martínez et al., 2021; Kunig et al., 2021; Peterson & Liu, 2023). These massive libraries are efficiently constructed through combinatorial synthesis of chemical building blocks, or synthons, with each resulting molecule being assigned a DNA barcode (see Figure 1). DELs are then used in selection experiments where they are mixed with proteins of interest bound to a matrix. Multiple rounds of washing are conducted to remove any weak binders, and the DNA tags of surviving molecules are sequenced as a measure of binding affinity. Data generated through these experiments are intrinsically noisy with various sources of bias arising from the DEL synthesis and selection processes, suggesting that modern machine learning methods may be needed to learn signal from the data. Unfortunately, there is still a lack of large, publicly available DEL datasets and benchmarking tasks to drive this important research area.

The growing interest in utilizing DEL data for modeling is evidenced by the many recent efforts to advance this area (Iqbal et al., 2024; Blevins et al., 2024; Gu et al., 2024). One of the primary reasons for this interest is that selection experiments using DELs alleviate some of the data limitations typical of the field; most chemistry problems in the machine learning domain lack consistent and sufficiently high-quality labels. In particular, DELs can contain billions of compounds, and require fewer resources to run compared to more traditional high-throughput screens (Peterson & Liu, 2023). However, the process of DEL synthesis and selection introduces various sources of bias that can add noise to the observed data. For instance, DEL experiments measure the affinity of molecules while they are attached to a DNA barcode (hence, on-DNA binding). In contrast, during a real therapeutic campaign, drug-like molecules are tested without a DNA tag (off-DNA), meaning the DEL data is biased by the DNA and the molecule's attachment to it. Moreover, uncertainties in reaction yields and biases in polymerase chain reaction (PCR) amplification add additional noise to the process. These difficulties have fueled an increasing enthusiasm within the community for developing structured computational models to more effectively interpret the data signals.

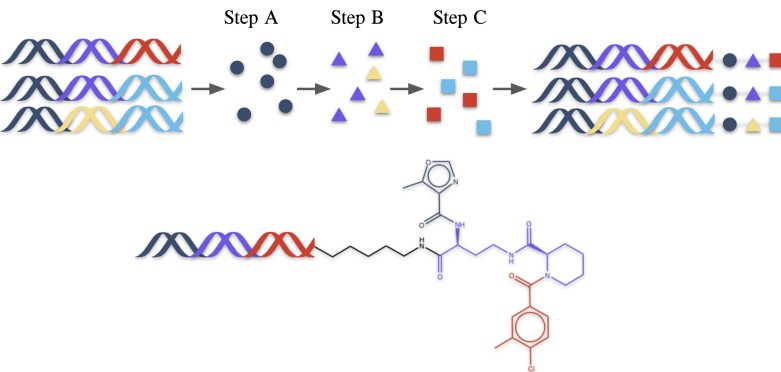

Figure 1: (Top) DELs are synthesized in a sequential manner; at every step, the DNA codon specifies the next building block (synthon) to be attached. (Bottom) An example of a fully synthesized DEL molecule, with the building blocks and codons colored-coded for visualizability. Typically, there is also a linker that connects the molecule to the DNA, which here is a 6-carbon chain.

Despite these challenges, several efforts have demonstrated successful applications of machine learning to DEL data (McCloskey et al., 2020). These early successes suggest that there is still much to explore in this domain. Currently, most methods have focused purely on discriminative modeling, which plays a crucial role in drug discovery campaigns by ranking prospective compounds in order to select potential hits. These prediction methods typically utilize established architectures, building upon both molecular fingerprints and more sophisticated graph convolutional networks (Duvenaud et al., 2015). More recently, efforts have been made to pose the problem in a probabilistic manner, directly incorporating experimental uncertainty in the model structure (Chen et al., 2024). These models leverage the fact that, while individual data points may be noisy, it is generally expected that groups of molecules with the same synthons or substructures will show enrichment, or signal, when analyzed collectively. Additionally, DEL data can be employed for generative modeling, providing weak supervision to navigate the complex chemical landscape, which is valuable for lead optimization steps of drug discovery.

To demonstrate the advantages of DEL data and promote development of the methods described above, we release **KinDEL** (**Kin**ase Inhibitor **D**NA-**E**ncoded **L**ibrary) as library of 81 million small molecules tested against two kinase targets, MAPK14 and DDR1. Our dataset, distinguished by its high consistency across experimental replicates (see Appendix B), provides a large amount of supervised data for the machine learning community to develop methods for solving small molecule chemistry problems in drug discovery.

In addition to the **KinDEL** dataset, we provide a set of benchmark tasks validated using biophysical assay data, which we also release publicly. A major challenge in driving research in this area has been the dearth of benchmark tasks to demonstrate the efficacy of using DEL data in deriving therapeutic insights. By releasing these benchmarks, we aim to facilitate the comparison of various modeling techniques currently applied to DELs. To seed these benchmarks, we survey computational methods from the literature and build predictive models, validated through biophysical data from compounds independently resynthesized both on- and off-DNA. Since DEL data primarily captures on-DNA binding events, but our interest lies in off-DNA binding affinity, these additional data are crucial for assessing the models' generalizability to diverse biophysical data. Our studies show that models built on DEL data can effectively characterize both on- and off-DNA affinities, highlighting the usefulness of DEL data in drug discovery. We hope that **KinDEL** serves as a public resource that can facilitate the iterative refinement of chemical models by providing supervised data in densely sampled chemical spaces.

## 2 DATASET

We first introduce a high level summary of the dataset generation, and then provide an overview of the data that we publicly release. The data generation is divided into roughly 3 experimental

processes, which are the synthesis of the DEL, the subsequent selection experiments against proteins of interest, and the biophysical assays to collect validation data. Specific experimental details can be found in Appendix A.

## 2.1 DATA GENERATION

**DEL Synthesis** The DEL is built as a trisynthon library with 378 synthons in the A position, 1128 synthons in the B position, and 191 synthons in the C position. The synthesis is a sequential process, with each synthon specified by the DNA tag and added one at a time. Notably, each molecule does not have only a single encoding; instead, multiple encodings of DNA map to the same final molecular structure. These redundant encodings help mitigate potential biases during the subsequent DNA sequencing step. Rather than counting the total number of amplified DNA associated with a single molecule (which is subject to PCR noise), we count the number of different redundant encodings observed for each molecule. This library was designed to enhance scaffold uniqueness and chemical diversity, thereby exploring a broader region of chemical space. A simplified diagram of synthesis can found in Figure 1.

**DEL Selection** DEL Selection was performed by combining the DEL with the target proteins, which were immobilized on beads, as well as a negative control without any protein. Multiple rounds of washes removed any weak binders in the solution. The molecules were then extracted via elution, and the subsequent samples were then amplified via PCR and sequenced to obtain the count data. Each selection was performed in triplicate with each protein. This process is visualized in Figure 2.

**Biophysical Assay Validation** Additional biophysical data on a small subset of molecules were collected both on- and off-DNA. For on-DNA data, we use Fluorescence Polarization (FP), which utilizes polarized light to quantify binding affinity by measuring the dissociation constant ($K_D$). For off-DNA data, we use Surface Plasmon Resonance (SPR), which also relies on light to measure the dissociation constant. The reason we collect both types of data is because on-DNA $K_D$ data reveals insights about the actual binding of the molecule in the DEL selection experiment, while off-DNA $K_D$ focuses on interactions of the molecules without a DNA-barcode, which is the relevant setting for an actual drug candidate.

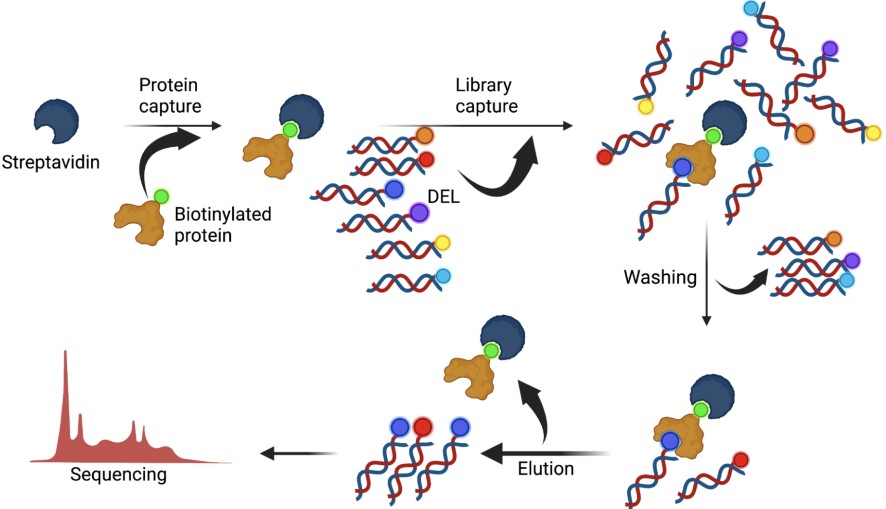

Figure 2: DEL selection experiments are conducted by combining the DEL with the protein target of interest immobilized onto Streptavidin beads. After multiple rounds of washing, the tight binders are eluted off of the protein, and their corresponding DNA are sequenced to obtain the count data.

## 2.2 DATA OVERVIEW

**DEL Data KinDEL** contains count data for more than 81M unique molecules used in selection experiments with two proteins MAPK14 and DDR1. Typically, DEL experiments are run with a

negative control without the protein, so that non-specific binding events (such as binding directly to bead) can be captured. In our dataset we provide three (3) replicates of data for each of the control and the protein target conditions. As mentioned earlier, raw DNA counts can be noisy due to PCR amplification bias (Aird et al., 2011; Kebschull & Zador, 2015), so we use the sequence count information as data, which measures the number of unique DNA sequences observed for each molecule.

**Pre-selection Data** Additionally, we provide data about the sequenced library itself, called pre-selection data, which provides a rough estimate of the relative abundance of each molecule prior to any experimental run. Due to the size of the library, it is too costly to sequence the library deeply enough for an extremely accurate pre-selection estimate. However, because there is always some amount of synthesis noise, for instance using impure reactants or having incomplete reaction yields, achieving a precise measurement of the pre-selection data is not imperative.

**Biophysical Assay Data** We have collected data from 30-50 molecules using the aforementioned biophysical assays, FP and SPR, to validate our models. For molecules on-DNA, we have re-synthesized molecules both from within and from outside our library. For off-DNA compounds, we have only resynthesized molecules that are within the DEL itself. These molecules were primarily selected from the top hits predicted from models trained on the DEL data (see Appendix C for more details).

Figure 3 illustrates various properties of the molecules in our dataset, comparing them to those of typical drug-like molecules. From this, we can see that **KinDEL** is well-posed within typical drug-like distributions according to an analysis by Shultz (2018). Notably, over 30% of the molecules in our library fall within the property ranges of already approved drugs, as outlined by Schultz. While certain synthon combinations may result in compounds that fall outside these preferred ranges, DEL molecules primarily serve to provide initial hits for drug discovery campaigns. These initial hits undergo iterative refinement during the hit-to-lead optimization process.

## 3 BENCHMARKING

### 3.1 EXPERIMENTAL SETUP

One primary use-case of DEL data is building predictive models of binding affinity. To that end, we investigate commonly used models in DEL literature as benchmark models and compare their performance in modeling binding affinity. In this benchmark, all models were trained using the top 1M compounds with the highest counts from our **KinDEL** dataset. We publish the full library to enable construction of further benchmarks.

**Held-out Test Set** The observed count data in DEL experiments are an approximation of the true on-DNA binding affinity ($K_D$). The count data are influenced by multiple sources of noise (see Section 4). We ultimately wish to rank molecules by binding affinity, so we use compounds with measured $K_D$ (from biophysical assays) as a test set. Performance on these compounds assesses if the models correctly rank compounds by $K_D$. This can be viewed as measuring how well models can remove the noise inherent to DELs.

For both our targets, MAPK14 and DDR1, the selected compounds contained in the DEL library were resynthesized on- and off-DNA to create an in-library held-out test set. For hit finding, we would like to be able to predict off-DNA $K_D$. This is challenging because the DEL data comes from DNA bound molecules, and is biased by the DNA. The on-DNA $K_D$ more closely aligns with DEL data since the molecules in the training data are bound to the same DNA in the same way. A few additional compounds were added from outside the library (and tagged with DNA) to create an additional held-out test set that we refer to as "Extended". The $K_D$ data from these biophysical assays are also released with our dataset. A UMAP visualization of the DEL including the in-library and external test set compounds is depicted in Figure 5b.

**Data Splits** We split our datasets using three strategies, ensuring that all held-out compounds are placed in the test set and not used for training. The first type of data split is a random split, where a randomly selected 10% of compounds are placed in the validation set, and another randomly selected 10% are placed in the test set. The second type of data split is a disynthon split, where we

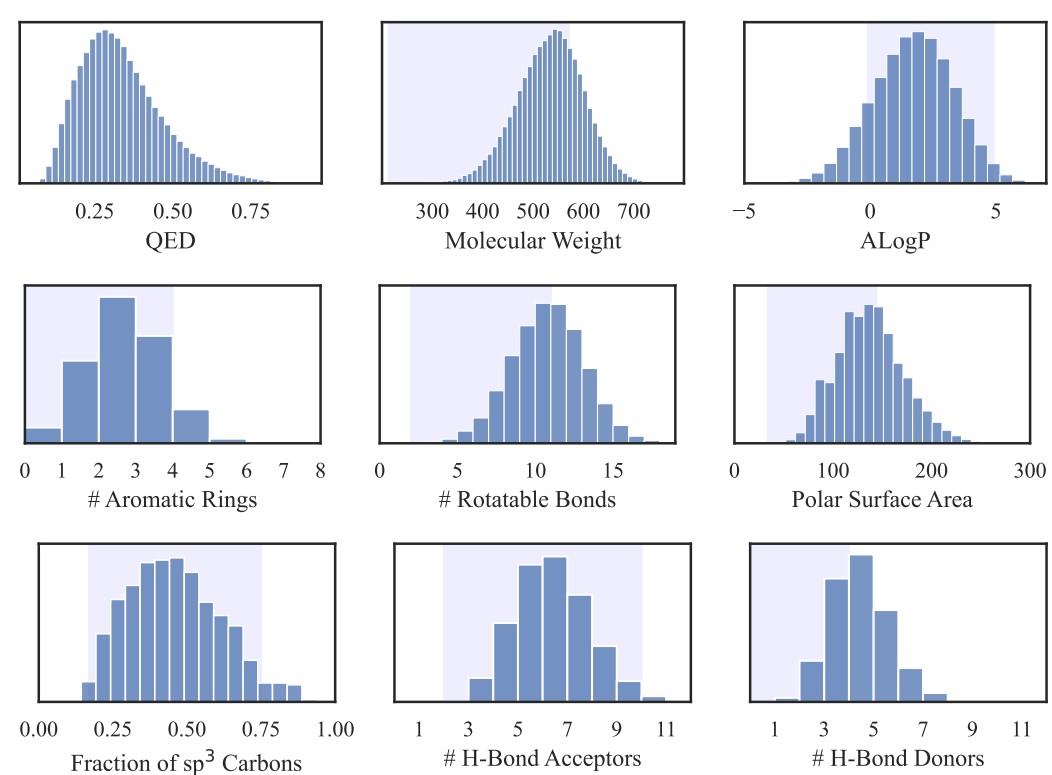

Figure 3: The distributions of chemical properties in the KinDEL dataset. These selected properties are often used to assess the druglikeness of molecules. The light blue areas mark the 10th and 90th percentiles computed for all the FDA approved oral new chemical entities, as reported by Shultz (2018). QED: quantitative estimate of druglikeness (Bickerton et al., 2012).

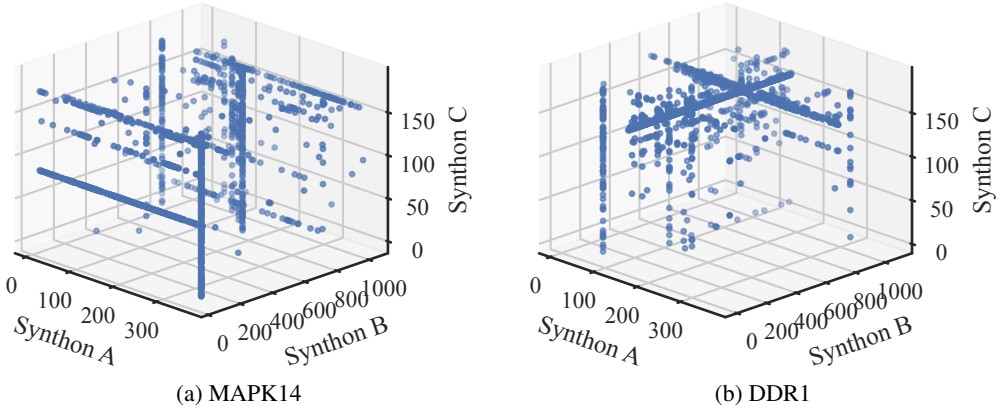

(a) MAPK14                    (b) DDR1

Figure 4: The 3D cube visualization of the dataset, where each axis corresponds to a different synthon in the DEL. Points in the plot are the most enriched compounds (measured using Poisson enrichment). The linear patterns can be interpreted as enriched disynthons, i.e. combinations of two synthons that often bind to the protein target.

(a) Experiment Test Setup

(b) KinDEL UMAP Visualization

Figure 5: **(a)** Testing data preparation includes the selection of the held-out testing compounds for the on- and off-DNA resynthesis (top) to perform binding assays and three types of data splits (bottom) used to prepare the internal testing set. **(b)** UMAP visualization of KinDEL constructed using Tanimoto distances between compounds. The compounds selected for the held-out testing set are depicted as orange diamonds (in-library) and green triangles (external).

sample disynthon structures (molecules with the same 2 synthons), and put all compounds containing these sampled structures in the same subset using the same 80-10-10 ratio between the training, validation, and test sets. This data partitioning method is more challenging for the machine-learning models because some synthon combinations tend to have a binding profile distinct from the individual synthons they consist of (see Figure 4). The third approach is a cluster split based on compound similarity. The details about the clustering algorithm can be found in Appendix E. The types of data splits are illustrated in Figure 5a. Each dataset is split five times for each splitting strategy, and the reported performance of the models is aggregated over five training runs.

**Evaluation Metrics** Models trained on DEL data typically predict an enrichment score, which can be regarded as a measure of binding affinity. To evaluate different baselines, we compare how well each method's predicted enrichment scores correlate with experimental $K_D$ values for the molecules in the held-out test set. Since different models are trained with different losses, this is a consistent way to compare model performance. Here, we use Spearman correlation, because we are primarily concerned with the ability of a model to rank order molecules by their binding affinity.

## 3.2 BENCHMARK MODELS

In this benchmark, we compare models commonly used for DEL data in the literature. These methods follow a similar paradigm, in which the models try to learn the protein binding signal in the target data by subtracting out noise from the control data. Two baselines are included to gauge the alignment in the actual data, between the DEL counts and validation $K_D$ values. The first baseline, count enrichment, computes an enrichment score by subtracting the average control counts from the average target counts. The second baseline, Poisson enrichment, computes a ratio of fitted Poisson distributions of the target and control data (Gerry et al., 2019).

Next, we compare ML models trained on the data to predict the aforementioned Poisson enrichment using a mean-squared error (MSE) loss. These models include: random forest (RF) (Breiman, 2001), XGBoost (Chen & Guestrin, 2016), k-nearest neighbors (kNN) (Fix & Hodges, 1989), and a deep neural network (DNN) using Morgan fingerprints (radius=2, length=2048) (Rogers & Hahn, 2010) from RDKit (Landrum, 2010) as input features. We also tested two molecular graph models,

Table 1: Model performance evaluation for **MAPK14**. The test loss column contains values of the loss function computed on test split. The performance for the on- and off-DNA "In-Library" is the negative Spearman correlation between model predictions and experimental $K_D$ for compounds resynthesized from the DEL. The performance for on-DNA "Extended" set includes additional compounds resynthesized on-DNA but not in the original DEL.

| | | | | In-Library | | Extended |
| | | | | on-DNA | off-DNA | on-DNA |
| | | | | Spearman's $\rho\uparrow$ | Spearman's $\rho\uparrow$ | Spearman's $\rho\uparrow$ |
| Split | Model | | Test Loss $\downarrow$ | $n=30$ | $n=33$ | $n=41$ |
|---|---|---|---|---|---|---|
| | Counts | | - | 0.778 | 0.353 | - |
| | Poisson | | - | 0.737 | 0.166 | - |
| random | RF | MSE | $0.064 \pm 0.003$ | $\mathbf{0.694 \pm 0.030}$ | $0.370 \pm 0.111$ | $0.453 \pm 0.028$ |
| | XGBoost | | $0.056 \pm 0.002$ | $0.477 \pm 0.009$ | $0.345 \pm 0.036$ | $0.196 \pm 0.074$ |
| | kNN | | $0.072 \pm 0.002$ | $0.649 \pm 0.041$ | $0.466 \pm 0.103$ | $0.464 \pm 0.040$ |
| | DNN | | $0.139 \pm 0.010$ | $0.582 \pm 0.062$ | $0.514 \pm 0.071$ | $0.351 \pm 0.058$ |
| | GIN | | $0.062 \pm 0.004$ | $0.511 \pm 0.038$ | $0.492 \pm 0.139$ | $0.174 \pm 0.067$ |
| | Chemprop | | $0.121 \pm 0.008$ | $0.693 \pm 0.039$ | $0.504 \pm 0.093$ | $0.462 \pm 0.063$ |
| | DEL-Compose$^{(M)}$ | NLL | $3.017 \pm 0.005$ | $0.448 \pm 0.054$ | $0.756 \pm 0.011$ | $\mathbf{0.569 \pm 0.048}$ |
| | DEL-Compose$^{(S)}$ | | $3.192 \pm 0.167$ | $0.420 \pm 0.050$ | $\mathbf{0.760 \pm 0.018}$ | - |
| cluster | RF | MSE | $0.063 \pm 0.015$ | $0.697 \pm 0.033$ | $0.313 \pm 0.071$ | $0.472 \pm 0.048$ |
| | XGBoost | | $0.059 \pm 0.013$ | $0.486 \pm 0.032$ | $0.378 \pm 0.091$ | $0.176 \pm 0.069$ |
| | kNN | | $0.080 \pm 0.018$ | $0.575 \pm 0.034$ | $0.435 \pm 0.094$ | $0.421 \pm 0.032$ |
| | DNN | | $0.065 \pm 0.011$ | $0.565 \pm 0.089$ | $0.592 \pm 0.065$ | $0.284 \pm 0.114$ |
| | GIN | | $0.072 \pm 0.015$ | $0.411 \pm 0.209$ | $0.369 \pm 0.054$ | $0.123 \pm 0.171$ |
| | Chemprop | | $0.131 \pm 0.047$ | $\mathbf{0.713 \pm 0.013}$ | $0.532 \pm 0.070$ | $0.485 \pm 0.036$ |
| | DEL-Compose$^{(M)}$ | NLL | $3.038 \pm 0.053$ | $0.440 \pm 0.045$ | $0.730 \pm 0.009$ | $\mathbf{0.583 \pm 0.032}$ |
| | DEL-Compose$^{(S)}$ | | $3.339 \pm 0.114$ | $0.369 \pm 0.050$ | $\mathbf{0.766 \pm 0.012}$ | - |
| disynthon | RF | MSE | $0.154 \pm 0.016$ | $0.157 \pm 0.138$ | $0.505 \pm 0.062$ | $0.302 \pm 0.096$ |
| | XGBoost | | $0.148 \pm 0.015$ | $0.377 \pm 0.054$ | $0.482 \pm 0.045$ | $0.212 \pm 0.126$ |
| | kNN | | $0.165 \pm 0.014$ | $\mathbf{0.402 \pm 0.074}$ | $0.266 \pm 0.078$ | $0.367 \pm 0.043$ |
| | DNN | | $0.160 \pm 0.017$ | $0.275 \pm 0.135$ | $0.429 \pm 0.118$ | $0.184 \pm 0.146$ |
| | GIN | | $0.153 \pm 0.011$ | $0.090 \pm 0.084$ | $0.483 \pm 0.151$ | $-0.080 \pm 0.071$ |
| | Chemprop | | $0.216 \pm 0.007$ | $0.390 \pm 0.091$ | $0.506 \pm 0.093$ | $0.228 \pm 0.121$ |
| | DEL-Compose$^{(M)}$ | NLL | $3.177 \pm 0.028$ | $0.120 \pm 0.070$ | $0.716 \pm 0.052$ | $\mathbf{0.421 \pm 0.054}$ |
| | DEL-Compose$^{(S)}$ | | $3.351 \pm 0.040$ | $0.128 \pm 0.049$ | $\mathbf{0.748 \pm 0.024}$ | - |

Graph Isomorphism Network (GIN) (Xu et al., 2018) and Chemprop DMPNN (Heid et al., 2023). DEL-Compose (Chen et al., 2024) is a probabilistic model that uses Morgan fingerprints as input and predicts the parameters of a zero-inflated Poisson distribution to maximize the likelihood of the observed count data. We train two variants of DEL-Compose, one with only full molecule structures (DEL-Compose$^{(M)}$), and one with synthon structures (DEL-Compose$^{(S)}$). The hyperparameters of all the models used in this study are presented in Appendix D.

The architectures of the neural network models follow the implementation in the original publications. The DNN architecture contains multiple linear layers with ReLU activation, batch normalization, and dropouts after each layer except for the last one (see Appendix D) . All neural networks were trained using the Adam optimizer until convergence with early stopping when the validation loss does not improve for more than 5 epochs.

### 3.3 BENCHMARK RESULTS

Tables 1 and 2 show the performance of various models on MAPK14 and DDR1, respectively. The Counts and Poisson enrichment baselines serve as an estimate of the alignment between DEL screening results and experimental $K_D$ computed directly from the sequence count data, measuring how well the DEL data itself predicts the $K_D$ in the follow-up assays. For DDR1, we see that DEL-Compose, which views the data from a probabilistic perspective, is the most performant model in all but the "Extended on-DNA" set. For MAPK14, relatively simple models (RF and kNN) perform best for the in-library on-DNA validation while DEL-Compose performs better off-DNA.

Table 2: Model performance evaluation for **DDR1**. The test loss column contains values of the loss function computed on test split. The performance for the on- and off-DNA "In-Library" is the negative Spearman correlation between model predictions and experimental $K_D$ for compounds resynthesized from the DEL. The performance for on-DNA "Extended" set includes additional compounds resynthesized on-DNA but not in the original DEL.

| | | | | In-Library | | Extended |
| | | | | on-DNA | off-DNA | on-DNA |
| | | | | Spearman's $\rho \uparrow$ | Spearman's $\rho \uparrow$ | Spearman's $\rho \uparrow$ |
| Split | Model | | Test Loss $\downarrow$ | $n = 39$ | $n = 49$ | $n = 54$ |
|---|---|---|---|---|---|---|
| | Counts | | - | 0.695 | 0.355 | - |
| | Poisson | | - | 0.779 | 0.441 | - |
| random | RF | MSE | $0.685 \pm 0.011$ | $0.578 \pm 0.034$ | $0.267 \pm 0.022$ | $0.608 \pm 0.021$ |
| | XGBoost | | $0.519 \pm 0.011$ | $0.553 \pm 0.032$ | $0.252 \pm 0.031$ | $0.587 \pm 0.025$ |
| | kNN | | $0.748 \pm 0.010$ | $0.599 \pm 0.025$ | $0.316 \pm 0.026$ | $0.508 \pm 0.036$ |
| | DNN | | $1.261 \pm 0.057$ | $0.703 \pm 0.025$ | $0.335 \pm 0.009$ | $0.668 \pm 0.033$ |
| | GIN | | $0.454 \pm 0.012$ | $0.572 \pm 0.044$ | $0.283 \pm 0.028$ | $0.579 \pm 0.037$ |
| | Chemprop | | $1.391 \pm 0.043$ | $0.729 \pm 0.017$ | $0.335 \pm 0.004$ | $\mathbf{0.680 \pm 0.021}$ |
| | DEL-Compose$^{(M)}$ | NLL | $2.873 \pm 0.003$ | $\mathbf{0.731 \pm 0.016}$ | $\mathbf{0.509 \pm 0.024}$ | $0.646 \pm 0.024$ |
| | DEL-Compose$^{(S)}$ | | $2.918 \pm 0.057$ | $0.689 \pm 0.048$ | $0.483 \pm 0.044$ | - |
| cluster | RF | MSE | $0.641 \pm 0.190$ | $0.586 \pm 0.025$ | $0.273 \pm 0.017$ | $0.615 \pm 0.019$ |
| | XGBoost | | $0.589 \pm 0.164$ | $0.569 \pm 0.024$ | $0.262 \pm 0.009$ | $0.599 \pm 0.013$ |
| | kNN | | $0.887 \pm 0.189$ | $0.581 \pm 0.061$ | $0.329 \pm 0.063$ | $0.489 \pm 0.049$ |
| | DNN | | $0.519 \pm 0.150$ | $0.708 \pm 0.015$ | $0.330 \pm 0.022$ | $0.673 \pm 0.014$ |
| | GIN | | $0.633 \pm 0.142$ | $0.524 \pm 0.081$ | $0.137 \pm 0.047$ | $0.599 \pm 0.047$ |
| | Chemprop | | $1.507 \pm 0.363$ | $0.732 \pm 0.028$ | $0.326 \pm 0.014$ | $\mathbf{0.690 \pm 0.023}$ |
| | DEL-Compose$^{(M)}$ | NLL | $2.891 \pm 0.079$ | $\mathbf{0.737 \pm 0.041}$ | $0.467 \pm 0.025$ | $0.540 \pm 0.037$ |
| | DEL-Compose$^{(S)}$ | | $2.993 \pm 0.065$ | $0.686 \pm 0.027$ | $\mathbf{0.482 \pm 0.019}$ | - |
| disynthon | RF | MSE | $1.151 \pm 0.151$ | $0.481 \pm 0.120$ | $0.330 \pm 0.081$ | $0.557 \pm 0.082$ |
| | XGBoost | | $0.989 \pm 0.131$ | $0.523 \pm 0.071$ | $0.241 \pm 0.031$ | $0.572 \pm 0.046$ |
| | kNN | | $1.109 \pm 0.088$ | $\mathbf{0.663 \pm 0.043}$ | $0.363 \pm 0.038$ | $0.523 \pm 0.036$ |
| | DNN | | $0.977 \pm 0.104$ | $0.572 \pm 0.063$ | $0.265 \pm 0.051$ | $\mathbf{0.598 \pm 0.055}$ |
| | GIN | | $0.966 \pm 0.090$ | $0.410 \pm 0.020$ | $0.070 \pm 0.031$ | $0.546 \pm 0.023$ |
| | Chemprop | | $1.690 \pm 0.192$ | $0.558 \pm 0.061$ | $0.310 \pm 0.038$ | $0.579 \pm 0.037$ |
| | DEL-Compose$^{(M)}$ | NLL | $3.184 \pm 0.025$ | $\mathbf{0.663 \pm 0.022}$ | $\mathbf{0.463 \pm 0.023}$ | $0.492 \pm 0.049$ |
| | DEL-Compose$^{(S)}$ | | $3.110 \pm 0.041$ | $0.563 \pm 0.084$ | $0.429 \pm 0.069$ | - |

Recall that off-DNA is the task more reflective of the setting for selecting actual drug candidates. Interestingly, we find that DEL-Compose rank orders the validation compounds off-DNA better than using enrichment metrics of the DEL data itself (Counts, Poisson). This is true for both targets, and on all three splits of the data. As mentioned earlier, DEL data is only indirectly correlated to off-DNA data, so this demonstrates that these structure-based models may have regularization properties that can denoise the DEL data. Most models perform similarly in the cluster split compared to the random split, indicating that the cluster split created based on fingerprint cluster is not too much more challenging. However, we observe that almost all models perform worse on the disynthon split compared to the random split. In particular, this change in performance is quite significant for the MAPK14 data, which might suggest that these models are overfitting to certain features. The disynthon split is a more challenging task, since we remove structures entirely from the training data, and the models have to infer based on chemical structures (out-of-distribution inference). Overall, these results demonstrate that models trained on DEL data can be used for hit selection, since models can predict enrichment scores that correlate well with on- and off-DNA biophysical data.

## 4 DISCUSSION

**Data Applicability and Limitations**    Our data and benchmarking tasks were designed to evaluate the ability to predict compounds within the DEL. Although we incorporated challenging splits of the data, such as disynthon splits, we have not evaluated the usefulness of this particular data for truly out-of-distribution sets. While we do include an extended held-out set with molecules from outside

of the DEL, this extended set is limited in size. We envision that models trained on this data can be used to make predictions for molecules from purchasable catalogues, but such applications will require further exploration, especially being cognizant of problems in this space such as domain of applicability (Weaver & Gleeson, 2008).

One limitation of DEL data is that it only measures on-DNA binding events, and on- and off-DNA binding are only loosely correlated (Hackler et al., 2019). While we have shown in our experiments that machine learning models can have nice regularization properties and learn some correlations to off-DNA binding, DEL data needs to be combined with additional 3-D structural data to fully understand these different binding modalities. To that end, we also publicly release docked 3-D poses of our library molecules to the target proteins to aid future model development by the community (see Appendix F). Additionally, DEL screens are sometimes run with an additional experimental condition to distinguish potential allosteric binders from orthosteric binders (Gironda-Martínez et al., 2021). This is usually achieved by running the experiment with a known inhibitor doped in at high concentrations. Because our data lacks this condition, it can be difficult to discern non-specific binding modes. We also recognize that one limitation of is dataset is that the two targets we provide are kinases. In order to make this dataset more broadly applicable for modeling DEL data, we additionally release our data on another well studied target, Bovine Carbonic Anhydrase (BCA), using the same library.

**Challenges and Future Directions**    DEL data is powerful in that it specifically densely samples particular chemical spaces, which can be leveraged to learn more powerful representations. However, DEL data suffers from experimental noise. In particular, there are unobserved factors such as synthesis noise that makes it difficult to separate out signal from noise in the data (Zhu et al., 2021). Additionally, since our observations are sequencing read counts rather than actual binding affinity, the measurements also suffer from PCR bias (Aird et al., 2011). While we have presented several benchmark methods that try to learn a denoised enrichment from structure-based models, how best to do this is still an open question in the field, and we hope that our dataset release will enable the development of more denoising methods.

Furthermore, our benchmark primarily focuses on building purely predictive models, which encapsulates most of the published work in this field currently. However, **KinDEL** holds significant potential for use in generative frameworks. Many generative modeling approaches in the small-molecule space lack sufficient supervised data to learn interesting sampling distributions. However, this is exactly the data that DEL provides, densely sampling around particular synthons or di-synthon structures. Therefore, we hope that DEL data combined with provided docking poses can be used to train or fine-tune generative models on specific protein targets.

## 5    RELATED WORK

### 5.1    CURRENT DATASETS

To the best of the authors' knowledge, only a limited number of DEL datasets have been released publicly. While **KinDEL** is not the largest library tested, our dataset is evaluated on two distinct targets in the Kinase family and contains comprehensive and consistently replicated raw experimental data (see Appendix B) in addition to orthogonal, non-DEL based binding affinity data for validation.

Iqbal et al. (2024) released three DEL datasets tested against two targets, Casein Kinase $1\alpha/\delta$. Their libraries span a range of chemical size and diversity, and they demonstrated the efficacy of a suite of machine learning methods to model binding affinities. Like our work, they experimentally validated some of their machine learning derived molecule predictions with biophysical assay data. However, they only tested off-DNA binding affinities, whereas we have synthesized our compounds both on- and off-DNA. Having both kinds of data is important because on-DNA data bridges the gap between DEL data and off-DNA data. Additionally, they have not investigated probabilistic models in their benchmarking tasks, which we find to be empirically useful.

Another recent dataset release is from Leash Bio, who released their data as a Kaggle competition (Blevins et al., 2024). Their dataset has a single DEL screened against on three different proteins, but their data has been preprocessed from raw count to a binary label. This process is often used to

denoise DEL data, but it removes information from the data. We show through our experiments that it is possible to learn over discrete count data through probabilistic approaches.

Gerry et al. (2019) and Hou et al. (2023) have released libraries tested against well-studied targets Horseradish Peroxidase and members of the Carbonic Anhydrase family. In both cases, the DELs were synthesized with specific chemotypes known to bind to their targets. As a result, evaluation in each is primarily limited to comparing relative enrichment for compounds containing known binders. These papers serve as excellent case-studies but with their targeted library construction and limited library size ~100k (Gerry et al., 2019) and ~7M (Hou et al., 2023), they are unlikely to serve as generalizable benchmarks for DEL ML methods.

## 5.2 Computational Methods on DEL Data

Computational efforts on DEL data have evolved over time. One of the major concerns with DEL data is its intrinsic noisiness. Firstly, the synthesis of a DEL is optimized for scale, and not precision, which results in uncertainty in the composition of the library. For instance, DEL synthesis will result in forming partial or truncated products (Gironda-Martínez et al., 2021; Binder et al., 2022), which has some causal effect on the observed data that cannot easily be measured. Additionally, the final data measurement is obtained by sequencing PCR-amplified DNA barcodes; however, the PCR process itself does not uniformly sample from the surviving members of the selection experiment (Aird et al., 2011).

Computational methodology for DEL data is still a nascent area, largely due to a lack of publicly available data. Current works typically directly apply fingerprint and graph neural network approaches that are popular within property prediction models. Due to the noisiness intrinsic to the data, many methods bin the labeled data into a binary classification to avoid overfitting on the raw data. For instance, McCloskey et al. (2020) trained classification models on disynthons using variants of typical graph neural networks. This approach was applied by Ahmad et al. (2023) to find first-in-class WD Repeat Domain 91 ligands, using molecules from buyable catalogues. Torng et al. (2023) follows up on this work and uses Weave GCN for hit identification of CA-IX. However, the true binding process is observed on a discrete scale, so more recent works have focused on building probabilistic models of binding. Lim et al. (2022) proposed a new uncertainty-based loss function for training regression models, and demonstrated the efficacy of their method using various graph neural network models. Shmilovich et al. (2023) takes this process one step further and incorporates 3-D docked poses to leverage the scale of DEL data by using multiple-instance learning to learn over poses. Other works exploit compositionality of DELs as inductive biases of the models. Binder et al. (2022) tries to explicitly model the partial products, while Chen et al. (2024) computes a hierarchical representation of DEL molecules and incorporates a probabilistic loss. Koziarski et al. (2024) explores generative models of DEL molecules using GFlowNets. Many of these developments are relatively new, and we hope that **KinDEL** will enable the further development of these methodologies in the field.

## 6 Conclusion

DELs have emerged as a high-throughput technology that enables screenings of large combinatorial small molecule libraries. However, DEL data has many sources of intrinsic noise stemming from synthesis and selection experiments, necessitating the right machine learning tools to extract the correct signal in the data. We introduce **KinDEL** as a 81M molecule dataset with selection data from two targets, MAPK14 and DDR1, in order to highlight the ability to model DEL data to find potent binders. Our DEL is built with chemical diversity in mind, and many of the molecules in the library have properties within the range of approved drugs. We additionally release biophysical data for molecules synthesized both on- and off-DNA to validate our models trained on DEL data. We hope that our public data release and benchmarking will engender more interest in DEL data as an important chemical modality for machine learning research in the future.

## 7 REPRODUCIBILITY STATEMENT

We have included both experimental details of the data generation as well as all the model details for our benchmarking tasks in Appendix A and D. The code to replicate our experiments and data can be found at: `https://kin-del-2024.s3.us-west-2.amazonaws.com/kindel.zip`. See the README file in the code for more details.

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

## A    EXPERIMENTAL PROTOCOLS

### A.1    DEL SYNTHESIS

**Library Design** The design of DNA-encoded libraries (DELs) often grapples with limited diversity and availability of bifunctional building blocks compared to their monofunctional counterparts. In order to expand structural diversity of the first two synthons of a three-step linear library, we implemented a hybrid design that combines trifunctional and monofunctional building blocks within one step of synthesis, to form a bifunctional synthon, by using DEL synthesis on solid supports. This strategy significantly expands the chemical space and diversity of our DELs, offering a robust pathway to discovering novel compounds with enhanced biological activity.

**Library Build** The DEL is built as a trisynthon library, comprised of 378 synthons in the A position, 1128 synthons in the B position (the terminal, capping step), and 191 synthons in the C position. This resulted in a DNA encoded library comprised of roughly 81 million unique members. The first two steps are done either by acylation with N-protected (Boc or Fmoc) amino acid, followed by deprotection, or by immobilization of the DNA to a solid support (DEAE sepharose resin) followed by a series of chemical transformations: acylation with trifunctional building block (Boc-amino acid with protected side chain: Fmoc-amine, ester or ketone), deprotection (if necessary) of the side chain protecting group, reaction (amide coupling, reductive amination/alkylation) with the monofunctional building block (amine, aldehyde or acid), and lastly eluting the DNA off the DEAE sepharose and cleavage of the Boc group. In the final step, the downstream amino groups were reacted with monofunctional acids or aldehydes. Each of the steps described herein is encoded by the attached DNA strand allowing for ready deconvolution of the sequence of chemical steps taken for any given member.

### A.2    DEL SELECTION

The DEL selection was performed using an Agilent Bravo for all handling steps. To immobilize the protein 1 nanomole of the protein of interest (Avi-tagged MAPK14 or DDR1) . were incubated through gentle aspiration through streptavidin-coated beads housed within PhyTips (Cat. #PTV-92-20-05). This was performed in technical triplicate for each protein. Following immobilization, any unbound proteins were washed away using Buffer A (45 mM HEPES, 45 mM Tris-HCl, 150 mM NaCl, 5 mM $MgCl_2$, pH 8.1).

Subsequently, the immobilized proteins were incubated through gentle aspiration and dispension over a period of 30 minutes with the Naive DEL library (at 500K copies of each member) in solution of Buffer A supplemented with 0.1 mg/ml sheared salmon sperm DNA to minimize non-specific binding. To eliminate noise and unbound library compounds, the Phynexus tips were washed six independent times with Buffer B (45 mM HEPES, 45 mM Tris-HCl, 425 mM NaCl, 5 mM $MgCl_2$, pH 8.1, 0.03% Tween).

Post-washing procedure. Any compounds remaining (the binders) were eluted by 5x aspiration through the tip of hot water at 90°C. The material eluted is termed 'Elution Round 1'. Following this the used Phytips were disposed, and fresh streptavidin PhyTip were used to capture a fresh 1 nanomole of avi-tagged protein of interest, and the process described above repeated, but, the Naive DEL Library is instead the 90% of the eluent from Round 1 (Elution Round 1) as the source of library. This procedure is repeated a final time with the Elution Round 2 sample to generate an Elution Round 3 sample.

From the 10% of eluates conserved from round 2 and the round 3 eluate were taken 5 μL of sample that was subsequently PCR amplified in 50 μL reactions using New England BioLabs 2x Q5 mastermix (Cat. #M0492S) and custom designed barcode primers to differentiate both the round, target and experiment. Following this 1 μL of the amplified material is transferred to a 25 μL PCR reaction with Q5 and custom P5 and P7 primers to install the relevant sequencing primers. The PCR mixture is purified using a QIAquick PCR Purification Kit (Cat. #28104), this sample is then further purified to remove any DNA products of incorrect size using a PippenHT. The purified samples are submitted to next-generation sequencing utilizing the Novaseq platform with a 2x150 BP S4 kit. The sequencing depth (number of read counts/sequence counts) is 514.5M/65.8M for MAPK14, 296.5M/49.8M for DDR1, and 440.7M/52M for BCA in all three replicates.

## A.3 Biophysical Assays

### A.3.1 Molecular Biology

hMAPK14-1 (1-360, WT) gene was cloned into pET-28a expression vector (Novagen) by polymerase chain reaction (PCR). An N-terminal His-TEV-Avi tag was added by using the forward primer 5'-CTTTAAGAAGGAGATATACCATGGGCCATCATCACCATCACCAC-3' and the reverse primer 5'-TTCGTGCCATTCAATTTTCTG-3'. The forward primer included the restriction site NcoI. The hMAPK14 (1-360, WT) gene was cloned using the forward primer 5'-CTCAGAAAATTGAATGGCACGAAATGTCTCAGGAGAGGCCCACG-3' and the reverse primer 5'-GTGGTGGTGGTGGTGGTGCTCGAGTTAGGACTCCATCTCTTCTTGGTCA-3'. The reverse primer included the restriction site XhoI. This generated plasmid pET28a-His-TEV-avi-hMAPK14-1 (1-360, WT).

hMAPK14-2 (1-360, WT) gene was cloned into pET-28a expression vector (Novagen) by PCR. An N-terminal His-TEV tag was added by using the forward primer 5'-CTTTAAGAAGGAGATATACCATGGGCCATCATCACCATCACCAC-3' and the reverse primer 5'-GCCCTGGAAGTACAGGTTCTC-3'. The forward primer included the restriction site NcoI. The hMAPK14 (1-360, WT) gene was cloned using the forward primer 5'-GAGAACCTGTACTTCCAGGGCATGTCTCAGGAGAGGCCCACG-3' and the reverse primer 5'-GTGGTGGTGGTGGTGGTGCTCGAGTTAGGACTCCATCTCTTCTTGGTCA-3'. The reverse primer included the restriction site XhoI. This generated plasmid pET28a-His-TEV-hMAPK14-2 (1-360, WT).

hDDR1-1 (593-913, $\Delta$730-735) gene was cloned into pFastBac1 expression vector by PCR. An N-terminal His-TEV tag was added by using the forward primer 5'-GGTCCGAAGCGCGCGGAATTCACCATGCACCATCACCATCACCAC-3' and the reverse primer 5'-ACCTTGGAAGTACAGGTTCTC-3'. The forward primer included the restriction site EcoRI. The hDDR1-1 (593-913, $\Delta$ 730-735) gene was cloned using the forward primer 5'-GAGAACCTGTACTTCCAAGGTCCTGGTGCTGTGGGTGACGGT-3' and the reverse primer 5'-GGCTCTAGATTCGAAAGCGGCCGCTTACACAGTGTTCAGAGCGTC-3'. The reverse primer included the restriction site NotI. This generated plasmid pFastBac1-His-TEV-hDDR1-1 [593-913 ($\Delta$730-735), WT].

hDDR1-2 (593-913, $\Delta$730-735) gene was cloned into pFastBac1 expression vector by PCR. An N-terminal His-TEV-Avi tag was added by using the forward primer 5'-GGTCCGAAGCGCGCGGAATTCACCATGCATCATCACCATCACCAC-3' and the reverse primer 5'-TTCGTGCCATTCAATTTTCTG-3'. The forward primer included the restriction site EcoRI. The hDDR1-2 (593-913, $\Delta$730-735) gene was cloned using the forward primer 5'-CTCAGAAAATTGAATGGCACGAACCTGGTGCTGTGGGTGACGGT-3' and the reverse primer 5'-GGCTCTAGATTCGAAAGCGGCCGCTTACACAGTGTTCAGAGCGTC-3'. The reverse primer included the restriction site NotI. This generated plasmidpFastBac1-His-TEV-Avi-hDDR1-2 [593-913 ($\Delta$730-735), WT].

BCA2-1 (1-260, WT) gene was cloned into pET28a expression vector by PCR. An N-terminal His-TEV-Avi tag was added by using the forward primer 5'-CTTTAAGAAGGAGATATACCATGGGCCATCATCACCATCACCAC-3' and the reverse primer 5'-TTCGTGCCATTCAATTTTCTG-3'. The forward primer included the restriction site NcoI. The BCA2-1 (1-260, WT) gene was cloned using the forward primer 5'-CTCAGAAAATTGAATGGCACGAAATGAGCCATCATTGGGGCTATGGCAAAC-3' and the reverse primer 5'-GTGGTGGTGGTGGTGGTGCTCGAGTTATTTCGGAAAGCCGCGCACT-3'. The reverse primer included the restriction site XhoI. This generated plasmid pET28a-His-TEV-Avi-BCA2(1-260,WT).

All cloning was performed using the Beyotime Seamless Cloning kit (D7010M). All plasmids were amplified using DH5$\alpha$ cells followed by DNA extraction. Insertion of the genes were verified by sequencing.

A.3.2 PROTEIN PRODUCTION

**MAPK14**

MAPK14 plasmid [either pET28a-His-TEV-avi-hMAPK14-1 (1-360, WT) or pET28a-His-TEV-hMAPK14-2 (1-360, WT)] was transformed into BL21-Gold (DE3) competent cells (Agilent, 230132) and plated on LB/agar/kanamycin (50 µg/mL) medium then left to grow at 37°C. Fresh colonies from transformed BL21-Gold cells were picked and used to inoculate 100 mL of LB medium (10 g tryptone, 10 g NaCl, 5 g yeast extract per liter water) supplemented with (50 µg/mL) kanamycin and cultured overnight at 250 rpm 37°C. The overnight culture was added to 2 L LB/kanamycin ((50 µg/mL)) and grown at 250 rpm 37°C until OD600 reached 0.600. Isopropyl ß-D-1-thiogalactopyranoside (IPTG) was then added (final concentration 0.3 mM) and the culture was left to continue to grow at 16°C, 250 rpm overnight.

Cells were harvested the following day at 8,000 rpm at 4°C and stored at -20°C. Cell pellets were resuspended in lysis buffer (50 mM Tris, 500 mM NaCl, 1 mM TCEP, 10% glycerol, pH 8.0 and 0.5 µL Benzonase) and lysed using a high pressure cell homogenizer. Lysate was centrifuged at 13,000 rpm for 30 minutes at 4°C, twice, to remove cell debris. Lysate supernatant was then applied to 6 mL of Ni-Resin (which had been pre-equilibrated with lysis buffer) and the lysate/Ni mixture was incubated for 1 hr at 4°C. Resin was then loaded into a column and the column was washed with wash buffer (50 mM Tris, 500 mM NaCl, 1 mM TCEP, 10% glycerol, pH 8.0) and eluted with elution buffer (50 mM Tris, 500 mM NaCl, 1 mM TCEP, 10% glycerol, pH 8.0 and 250 mM imidazole). Eluted protein was cleaved with TEV protease at a 1:10 (w/w) ratio and dialyzed against dialysis buffer (50 mM Tris, 100 mM NaCl, 1 mM TCEP, 7.5 mM MgCl2, pH 8.0) at 4°C overnight.

Protein was loaded onto a Ni-resin column, pre-equilibrated with lysis buffer and the cleaved protein was collected in the flow-through (cleaved-tag left on the column). In the case of Avi-tagged protein (MAPK14-1), the dialyzed protein was incubated at 18°C for 4 hrs with ATP (1 mM), Biotin (0.5 mM), and BirA (200 nM), prior to loading onto Ni-resin to remove cleaved hexahistidine tags.

Protein was collected and concentrated with a 10 kDa Millipore Amicon Ultra-15 centrifugal filter unit. Concentrated protein was loaded onto a Superdex 200 column pre-equilibrated with SEC buffer. Protein was collected according to UV-vis signal and follow-up SDS-PAGE verification. Protein fractions were pooled and concentrated with a 10 kDa Millipore Amicon Ultra-15 centrifugal filter unit and exchanged into a final buffer containing 10 mM HEPES, 200 mM NaCl, 1 mM TCEP at pH 7.5. A final purified protein yield of 19 mg/L cell culture was obtained.

**DDR1**

DDR1 plasmid (either pFastBac1-His-TEV-Avi-hDDR1-2 [593-913 (Δ730-735), WT] or pFastBac1-His-TEV-hDDR1-1 [593-913 (Δ730-735), WT]) was transformed into DH10Bac competent cells (Agilent) and plated on LB/agar plates containing 50 µg/mL kanamycin, 7 µg/mL gentamicin, 10 µg/mL tetracycline, 100 µg/mL X-gal and 40 µg/mL IPTG then left to grow at 37°C for 48 hours. Fresh colonies from transformed DH10Bac competent cells were picked and used to inoculate 5 mL of LB medium (10 g tryptone, 10 g NaCl, 5 g yeast extract per liter water) supplemented with 50 µg/mL kanamycin, 7 µg/mL gentamicin, 10 µg/mL tetracycline and cultured overnight at 37°C. An aliquot of each cell culture was verified to contain the recombinant bacmid by PCR analysis.

Transfection. In a 6-well plate, Sf9 cells were grown in 2 mL cultures to a cell density of 2 x 106 cells/mL in SF900II medium at 27°C. The culture medium was then exchanged with fresh SF900II medium followed by inoculation with the recombinant bacmid:lipid (Cellfectin II reagent) complexes. The bacmid infected cells were incubated at 27°C for 5 days. The cell culture medium was collected as P1 virus and cell pellet was used for western blot analysis.

P2 baculovirus generation and scale-up protein expression. Sf9 cells were grown in a 50 mL culture medium to a cell density of 2 x 106 cells/mL followed by infection with P1 virus in the ratio 1:200. Incubate the P1 infected Sf9 cells at 27°C for 3 days. The cell culture medium was collected as P2 virus and the cell pellet from 1 mL cell culture was used for western blot analysis. For scale-up expression, Sf9 cells were grown in a 12 L culture medium to a cell density of 2 x 106 cells/mL followed by infection with P2 virus in the ratio 1:200. For N-Avi tagged hDDR1-2, BirA was co-expressed (infection ratio 1:500, biotin 40 µM) with the P2 infected Sf9 cells. The P2 infected Sf9 cells were incubated at 27°C for 3 days.

Cells were harvested after 3 days at 8,000 rpm at 4°C and stored at -20°C. Cell pellets were resuspended in lysis buffer (50 mM HEPES, 300 mM NaCl, 2 mM TCEP, 10 mM MgCl$_2$, 10% saccharose, cocktail, 100U/mL Benzonase, pH 8.0) and Protease Inhibitor Cocktail Tablet was added until a final concentration of 1 tablet/L. Cells were sonicated to lyse in repeating periods of 3 seconds on, 3 seconds off for 10 minutes. This cycle was then repeated for an additional 10 minutes. A color change of cell lysate was used to help indicate if cell lysis was sufficient, if no color change from pre-lysed cells was observed, cell lysis was allowed to continue. Cell lysate was centrifuged at 13,000 rpm for 30 minutes at 4°C, twice, to remove cell debris. Lysate supernatant was then applied to 6 mL of Ni-Resin (which had been pre-equilibrated with lysis buffer) and the lysate/Ni resin mixture was incubated for 1 hr at 4°C. Resin was then loaded into a column and the column was washed with wash buffer (50 mM HEPES, 300 mM NaCl, 2 mM TCEP, 10 mM MgCl$_2$, pH 8.0) and eluted with elution buffer (50 mM HEPES, 300 mM NaCl, 2 mM TCEP, 10mM MgCl$_2$, pH 8.0, 50 mM imidazole). Eluted protein was cleaved with TEV protease at a 1:10 (w/w) ratio and dialyzed against dialysis buffer (50 mM HEPES, 300 mM NaCl, 2 mM TCEP, 10 mM MgCl$_2$, pH 8.0) at 4°C overnight.

Protein was loaded onto a Ni-resin column, pre-equilibrated with lysis buffer and the cleaved protein was collected in the flow-through (cleaved-tag left on the column). Protein was collected and concentrated with a 30 kDa Millipore Amicon Ultra-15 centrifugal filter unit. Concentrated protein was loaded onto a Superdex 200 column pre-equilibrated with SEC buffer. Protein was collected according to UV-vis signal and follow-up SDS-PAGE verification. Protein fractions were pooled and concentrated with a 30 kDa Millipore Amicon Ultra-15 centrifugal filter unit and exchanged into a final buffer containing 20 mM HEPES, 200 mM NaCl, 1 mM TCEP, 5% glycerol at pH 7.5. A final purified protein yield of 1.8 mg/L cell culture was obtained.

**BCA2**

The BCA plasmid, pET28a-His-TEV-Avi-BCA2(1-260,WT), was transformed into BL21-Gold (DE3) competent cells (Agilent, 230132) and plated on LB/agar/kanamycin (50 μg/mL) medium then left to grow at 37°C. Fresh colonies from transformed BL21-Gold cells were picked and used to inoculate 100 mL of LB medium (10 g tryptone, 10 g NaCl, 5 g yeast extract per liter water) supplemented with 50 μg/mL kanamycin and cultured overnight at 220 rpm 37°C. The overnight culture was added to 10 L LB/kanamycin (50 μg/mL) and grown at 180 rpm 37°C until OD600 reached 0.600. Isopropyl ß-D-1-thiogalactopyranoside (IPTG) was then added (final concentration 0.3 mM) and the culture was left to continue to grow at 16°C, 160 rpm overnight.

Cells were harvested the following day at 8,000 rpm at 4°C and stored at -20°C. Cell pellets were resuspended in lysis buffer (50 mM Tris, 500 mM NaCl, 1 mM TCEP, 1 mM PMSF, 10% glycerol, pH 8.0 and 100 U/mL Benzonase) and lysed using a high pressure cell homogenizer. Lysate was centrifuged at 13,000 rpm for 30 minutes at 4°C, twice, to remove cell debris. Lysate supernatant was then applied to 5 mL of Ni-Resin (which had been pre-equilibrated with lysis buffer) and the lysate/Ni mixture was incubated for 1 hr at 4°C. Resin was then loaded into a column and the column was washed with wash buffer (50 mM Tris, 500 mM NaCl, 1 mM TCEP, 10% glycerol, pH 8.0) and eluted with elution buffer (50 mM Tris, 500 mM NaCl, 1 mM TCEP, 10% glycerol, pH 8.0 and 250 mM imidazole). Eluted protein was cleaved with TEV protease at a 1:10 (w/w) ratio and dialyzed against dialysis buffer (50 mM Tris, 100 mM NaCl, 1 mM TCEP, 7.5 mM MgCl2, 5% glycerol, pH 8.0) at 4°C overnight.

The dialyzed protein was incubated at 18°C for 4 hrs with ATP (1 mM), Biotin (0.5 mM), and BirA (200 nM), prior to loading onto Ni-resin to remove cleaved hexahistidine tags. Protein was loaded onto a Ni-resin column, pre-equilibrated with lysis buffer and the cleaved protein was collected in the flow-through (cleaved-tag left on the column).

Protein was collected and concentrated with a 30 kDa Millipore Amicon Ultra-15 centrifugal filter unit. Concentrated protein was loaded onto a Superdex 75 column pre-equilibrated with SEC buffer. Protein was collected according to UV-vis signal and follow-up SDS-PAGE verification. Protein fractions were pooled and concentrated with a 30 kDa Millipore Amicon Ultra-15 centrifugal filter unit and exchanged into a final buffer containing 25 mM HEPES, 200 mM NaCl, 1 mM TCEP at pH 7.5. A final purified protein yield of 8.38 mg/L cell culture was obtained.

BCA used for fluorescence polarization assay was obtained from Sigma-Aldrich (Product No. C2522).

### A.3.3 Biophysical Methods

**Fluorescence Polarization**

Annealing of DNA tagged DEL molecules. DEL small molecule hits are attached to the DNA oligo, Za (GCAGGCGGAGACCTGCAGTCTG). Fluorescein (Integrated DNA Technologies, /3FluorT/) tagged complementary DNA oligo Za' (CAGACTGCAGGTCTCCGCCTG/3FluorT/) was annealed to Za-tagged DEL compounds using a thermocycler (Bio-Rad, 1851148).

Assay setup. In a 384 well black small volume microplate (Greiner Bio-One, 784900), the annealed compounds were dispensed using Echo650 (Labcyte) at a constant final concentration of 4 nM in 137 mM NaCl, 2.7 mM KCl, 9.8 mM Phosphate buffer, 0.01% Tween-20. MAPK14-2, DDR1-1, and BCA were each serially diluted 1:1 starting from a top dose of $50\,\mu$M in 137 mM NaCl, 2.7 mM KCl, 9.8 mM Phosphate buffer, 0.01% Tween-20 for a total 16-point dilution and were transferred to the 384 well black small volume microplate containing the annealed compounds using Agilent Bravo G5563A. The final assay concentration was 2 nM the annealed compound and $25\,\mu$M top dose of the protein. The final assay volume was $10\,\mu$L. The serially diluted protein was incubated with the annealed DEL compounds and then the fluorescence polarization was measured using PerkinElmer EnVision 2105 in milli-P (mP), where $mP = 1000 \times (S - GP)/(S + GP)$ ($S$ and $P$ are background subtracted fluorescence count rates, and $G$ is an instrument and assay dependent factor). The dose-response curves ($mP$ vs. [Protein]) were fit in GraphPad Prism using the equation, $Y = Bmax \times X/(K_D + X) + NS \times X + Background$, where $Y$ is $mP$, $X$ is [Protein], $Bmax$ is the maximum specific binding in the same units as $Y$, $K_D$ is the equilibrium dissociation constant, in the same units as $X$, $NS$ is the slope of nonspecific binding in $Y$ units divided by $X$ units, $Background$ is the amount of nonspecific binding with no added radioligand.

**Surface Plasmon Resonance**

Compounds were tested using Surface Plasmon Resonance (SPR) using a Biacore T200 and a Biacore S200 (Cytiva Life Sciences). N-terminally biotinylated DDR1 [593-913 ($\Delta$730-735)], MAPK14 (1-360), and BCA2-1 (1-260, WT) were immobilized to a streptavidin coated SPR chip (Series S SA, Cytiva, 29699621). Approximately 4400 RU of hDDR1-2, 4900 RU of MAPK14-1, and 3300 RU of BCA2-1 were immobilized for dose response assays using 1xHBS-P+ (10 mM HEPES, 150 mM NaCl, 0.05% v/v Surfactant P20; pH 7.4). 1xHBS-P+ was prepared by mixing 10xHBS-P+ (Cytiva, BR100671) with HPLC-grade water (Fisher Scientific, W51).

Compounds were diluted in the SPR running buffer which consisted of 1xHBS-P+ supplemented with 5% DMSO (VWR, 76177-938). Multi-cycle kinetics was used to determine compound affinities. Compounds were injected in a series of increasing concentrations with a $30\,\mu$L/min flow rate, a 60 s contact time and a 300 s dissociation time. Compound sensorgrams were processed and fit using Biacore Evaluation software, with DMSO correction applied during sensorgram processing. Binding and dissociation curves for each compound were globally fit to obtain an on- and off-rate kinetic constant ($k_a$ and $k_d$, respectively) which were used to determine the overall binding constant ($K_D$) for each compound.

## B  Experimental Replicability

DEL experiments suffer from the substantial noise that stems from the large scale of combinatorial libraries screened simultaneously within a single tube. Key sources of this noise include inconsistent control over the quantities of each compound in the mixture, synthesis-related challenges such as the introduction of side products and low synthetic yields, as well as errors during DNA sequencing. Figure 6 illustrates the strong correlation between experimental replicates in our DEL experiments, underscoring the reproducibility of the panning procedures we conducted.

## C  Validation Set Selection

We selected validation compounds to cover a range of $K_D$ values and chemical diversity, allowing us to better assess the models' ability to rank molecules across diverse binding affinities. To achieve this, we used an ensemble of models described in Section 3, and then clustered the molecules based on chemical similarity via Butina clustering using Morgan fingerprints. This constituted the

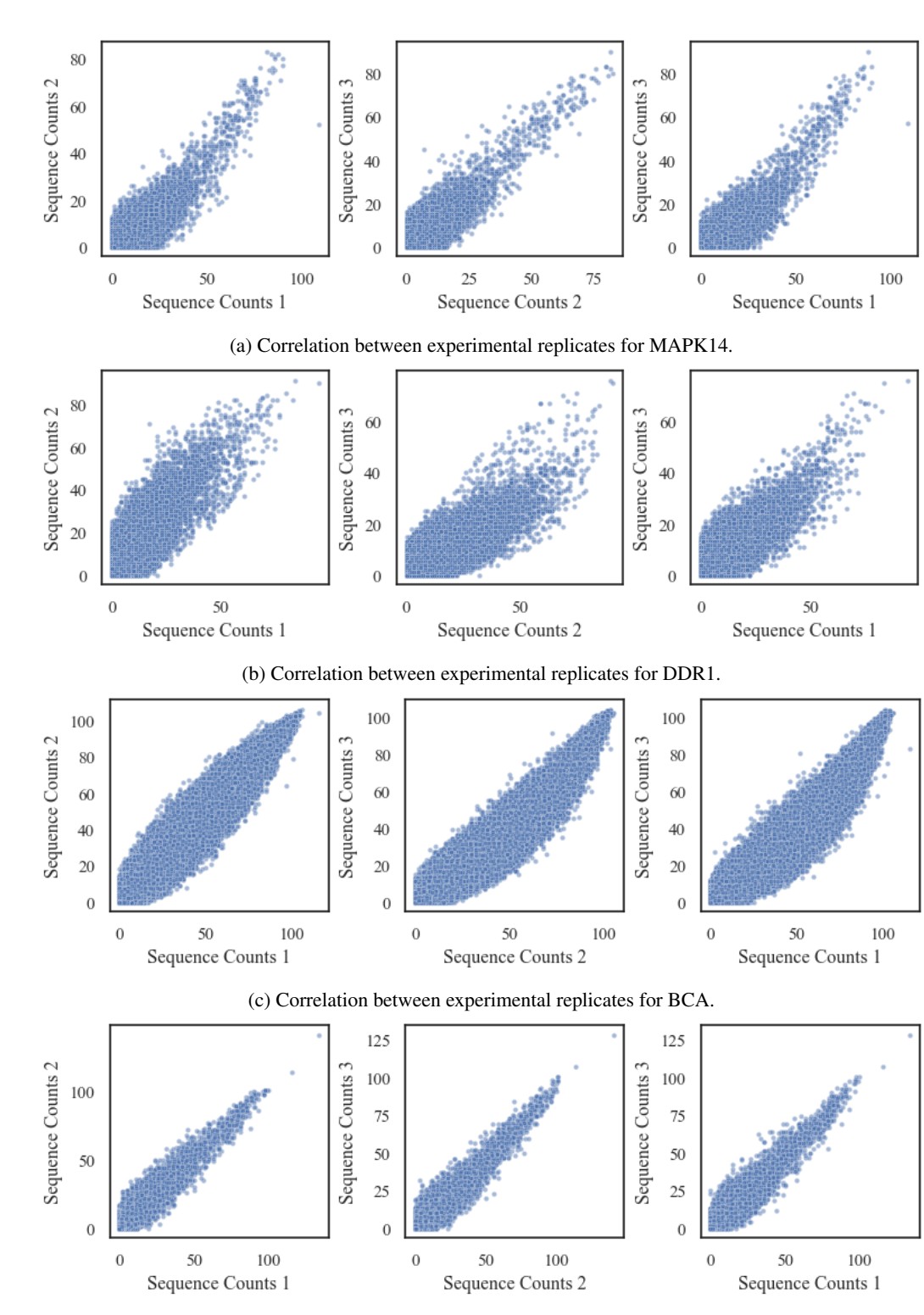

(a) Correlation between experimental replicates for MAPK14.

(b) Correlation between experimental replicates for DDR1.

(c) Correlation between experimental replicates for BCA.

(d) Correlation between experimental replicates for the control.

Figure 6: Correlation between experimental replicates. Binding to each target and the control was measured in three replicates, which are included in the published dataset.

molecules in the "in-library" set. We also surveyed literature for tool compounds, which we also re-synthesized on-DNA. This made up the additional molecules found in the "extended" set.

## D    MODEL HYPERPARAMETERS

This section describes the hyperparameters used to train the models presented in this study. Random Forest and XGBoost used 100 decision trees trained with the squared error criterion, and the depth of decision trees was not restricted. The k-Nearest Neighbors model used 5 nearest neighbors.

The final architecture of the DNN model consisted of 5 linear layers with the ReLU activation functions except for the last one. Batch norm and dropout layers (with the probability of zeroing an element equal to 20%) were applied after each layer before the activation layers. The hidden dimension size was set to 512 for all layers.

The GIN model has 5 GIN convolutional layers with hidden dimension size equal to 256. The graph convolutional layers are followed by the global average pooling layer and two linear layers with a ReLU activation layer between them. The first linear layer reduces the dimensionality from 256 to 128 before the second layer produces the model prediction.

Chemprop uses three layers of bond message passing with the hidden dimension of 300. Then, the information is aggregated using a global average pooling layer, and one linear layer is used to make predictions. The model is trained using the default Chemprop optimization procedure, which employs the Adam optimizer with a Noam learning rate scheduler and 2 epochs of warmup.

The DEL-Compose model was used in two modes of molecule encoding: synthon-based encoding (denoted DEL-Compose$^{(S)}$) and full molecule encoding (denoted DEL-Compose$^{(M)}$). The full molecule encoding uses four linear layers with ReLU activation functions to learn the encoding of the molecule based on its Morgan fingerprint. The synthon-based mode embeds each synthon separately using the same 4-layer MLP. Next, combinations of synthon embeddings (AB, BC, AC, ABC) are further processed by 2-layer MLPs, and finally all embeddings are aggregated using a 4-head attention pooling. The output distribution was set to zero-inflated Poisson distribution. The learning rate used to train DEL-Compose was 5e-5, and the batch size was 64.

## E    DATASET SPLITTING

It is often difficult ensure that information from the test set does not leak into train. One popular method is calculating Bemis-Murcko scaffolds (Bemis & Murcko, 1996) for each compound in a dataset and assigning compounds to a split by scaffold. Unfortunately, this method often fails for DEL data. Below we calculate Bemis-Murcko scaffolds for the top million compounds in the MAPK14 dataset using rdkit (Landrum, 2010). There are ∼300,000 unique scaffolds calculated for these compounds, implying many compounds have a unique scaffold (see Fig. 7a). Additionally the common scaffolds are often trivial (see Fig. 7b). The most common scaffold is a simple benzene, and the next five most common are also extremely common molecular building blocks. Given these shortcomings, we do not use a scaffold split.

An alternative method is a similarity split based on clustering with fingerprints. A classic method for this is Butina Clustering (Butina, 1999). Unfortunately, it does not scale well to large datasets. We have developed an internal method that scales to large datasets. Our two step method first uses UMAP (McInnes et al., 2018) to reduce 1024 ECFP4 fingerprints (Rogers & Hahn, 2010) to length 10 float vectors. Then HDBSCAN (McInnes et al., 2017) is used to cluster compounds, in a GPU implementation from NVIDIA (cuml) this runs in a few hours on a single Tesla T4. Clusters are assigned to splits using a waterfall method, at each step assigning the largest remaining cluster to the smallest current split. The following settings were used:

```
UMAP(n_components=10, metric="jaccard", n_neighbors=30,
min_dist=0.0, n_epochs=1000)

HDBSCAN(min_cluster_size=10, min_samples=None, metric='euclidian',
prediction_data=True, cluster_selection_method='eom')
```

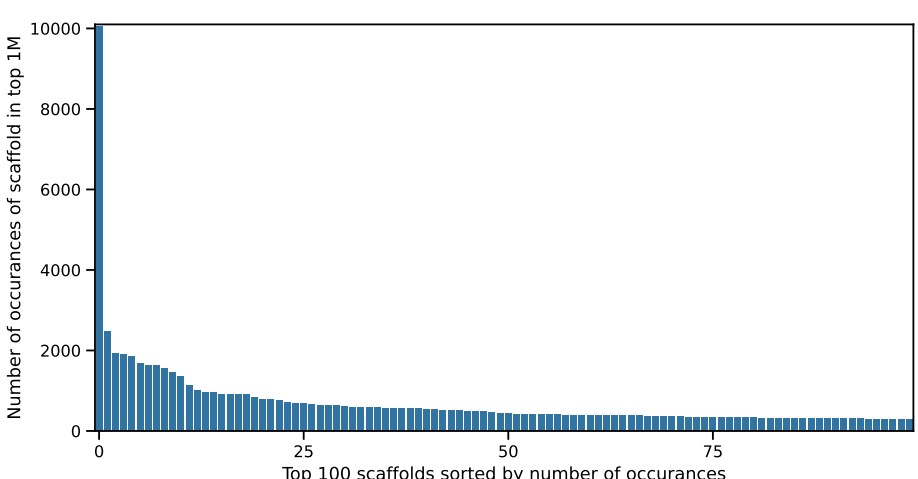

(a) Number of occurrences for the 100 most common scaffolds in the top one million compounds for MAPK14. The number of compounds assigned to each scaffold declines rapidly, and many scaffolds are unique.

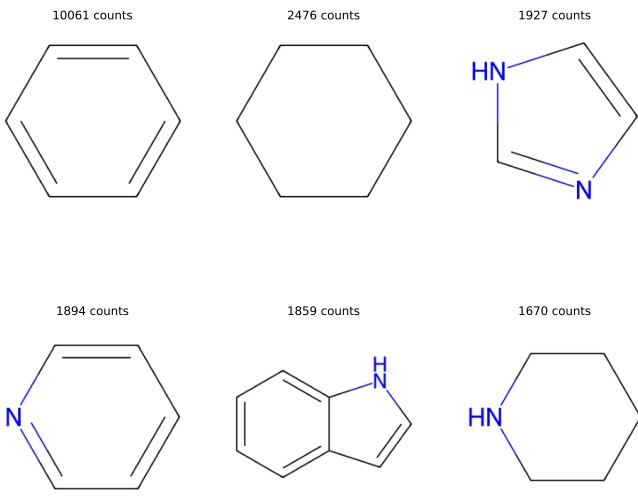

(b) The top six most frequent scaffolds are common structures that are not pharmacologically "interesting".

Figure 7: Bemis-Murcko scaffolds in KinDEL. The high number of unique scaffolds and the simplicity of the most common scaffolds indicate that a scaffold-based split is not challenging enough for our benchmark.

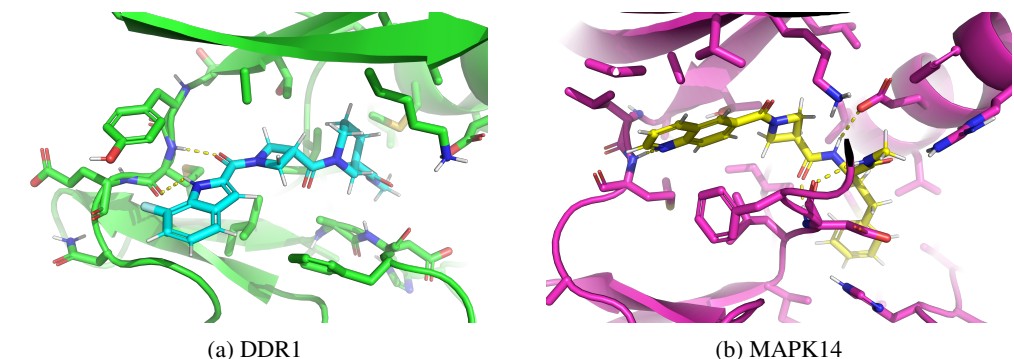

(a) DDR1            (b) MAPK14

Figure 8: Example ligands from the KinDEL datasets docked to our selected kinases; **(a)** example pose of a library member docked to a DFG-out conformation of DDR1 (PDB: 6FEX), forming extensive hinge interactions. Beta-1 and -2 hidden for clarity; **(b)** example pose of a library member docked to a DFG-out conformation of MAPK14 (PDB: 3KQ7), bridging the activation loop and C-alpha glutamate.

## F  MOLECULAR DOCKING PROCEDURE

Ligand stereoisomer enumeration (up to 8), tautomer or protonation state selection, and coordinate generation were performed with LigPrep from Schrödinger Suite 2024-4, using Epik 7 (Johnston et al., 2023) and OPLS4 force field (Lu et al., 2021) at pH 7.4. Docking was performed using the Vina scoring function (Trott & Olson, 2010) in Uni-Dock 1.1.2 (Yu et al., 2023) with `exhaustiveness` 512 and `max_step` 60, saving the top three poses per receptor. All of these 234k ligand states were docked against all six receptor models, using a fixed 20 Å docking box centered on the orthosteric pocket.

One receptor model represents the major conformation of DDR1 in available experimental structures, the DFG-out, C-helix-in PDB: 6FEX (Richter et al., 2018). Five receptor models were chosen to represent most experimental structures of MAPK14, with a variety of activation loop and P-loop conformations, PDB: 3KQ7, 3S3I, 5WJJ, 5XYY, and 6SFI (Cheng et al., 2009; Aiguadé et al., 2012; Kaieda et al., 2018; Wang et al., 2017; Rohm et al., 2019). All receptors were prepared using the Protein Preparation Wizard from Schrödinger Suite 2023-4, capping termini with neutral ends, and aligned into a common frame.

Docked poses are provided in SDF format with "molecule_hash" and "receptor" properties. Example poses are shown in Figure 8, where the hinge binding motif that is characteristic for kinases can be observed.

