# OpenReview forum: "KinDEL: DNA-Encoded Library Dataset for Kinase Inhibitors"
_ICLR.cc/2025/Conference — Submitted to ICLR 2025_

### Official Review · Reviewer_63rq · 2024-10-18

**Soundness:** 4
**Presentation:** 3
**Contribution:** 3
**Rating:** 6
**Confidence:** 4

**Summary:**

This paper propose a new open-source dataset as well as related benchmark for the DEL community. The main motivation behind this paper are:

1. DEL community lacks a large, publicly available DEL dataset to benchmarking tasks.
2. Current DEL dataset contains large bias and noise, and the existing methods cannot greatly address this issue.

So, the authors propose an open-source dataset and a related enhancement approach to address these challenges. In details, these improvements and contributions are:

1. KinDEL: a library of 81 million small molecules tested against two kinase targets, MAPK14 and DDR1, which is novel and with large amount.
2. A comprehensive benchmark tested on current computational methods with on both on-DNA and off-DNA settings.

The proposed dataset and benchmark have great potentials to stimulate the development of the community.

**Strengths:**

1. The dataset constuction process is reasonable, sound, and comprehensive. Also, the related process is explained clearly.
2. The corresponding evaluation to the proposed dataset is comprehensive, and the two types of splits "random" and "disynthon" enhance the perspective of the benchmark.
3. The evaluation is clear and straightforward (Table1, Table2, and Figure6).
4. The comparison to the current datasets is well-written and shows the valuable insights from authors and their advanced understanding to the DEL-related tasks.

**Weaknesses:**

1. While there are very interesting performance comparison in the shown Table (Table1 and Table2), the explaination of the experiment results are expected, such as "why RF method performs the best on on-DNA set (Line 335)", "why there are different performance rankings on on-DNA and off-DNA settings?" and "why DEL-Compose(M) and DEL-Compose(S) performs differently on on-DNA and off-DNA settings?". I believe the insight provided by the authors would make the benchmark more solid and comprehensive.
2. The proposed experiments including general performance (Table1, and Table2), and the visualization of experimental replicates are somehow not comprehensive enough. Serving as a benchmark for the DEL-community, more views and new settings are required, such as case study of top-ranking candidates (which can be potential candidates for real-world application), subset-Spearman coefficient (which is proposed in the paper "DEL-Dock: Molecular Docking-Enabled Modeling of DNA-Encoded Libraries" [1]), and potential effect of chemical properties and data source selections (building blocks) to the method performances on the KinDEL dataset. Then, the multi-view perspective of the benchmark could lead to larger contribution to the community.

Reference:
1. Shmilovich K, Chen B, Karaletsos T, et al. DEL-Dock: Molecular Docking-Enabled Modeling of DNA-Encoded Libraries[J]. Journal of Chemical Information and Modeling, 2023, 63(9): 2719-2727.

**Questions:**

1. According to Table1 and Table2, the SOTA performance w.r.t. Spearman coefficient can reach over 0.7, which is a very promising result,  but in paper "DEL-Dock: Molecular Docking-Enabled Modeling of DNA-Encoded Libraries" [1], there's an another proposed dataset containing Fingerprint and docking poses, where existing methods can only achieve relatively poor performance (around 0.30 w.r.t. negative Spearman coefficient) compared to KinDEL dataset. May the authors explain what is the inner differeneces between two datasets that leads to the obvious differences between two datasets? I also believe this comparison can make this work more solid as the contributing benchmark.
2. Is it possible to provide an advanced version of KinDEL dataset with machine-aided of molecule docking poses. I understand it's very time-consuming and CPU-resource costly, but the DEL dataset with 1D and 3D modalities would lead to wider applications and evaluations to this community. Considering the potential cost, I believe it is also great to have dataset with only fingerprint (1D) information of the molecules.

Reference:
1. Shmilovich K, Chen B, Karaletsos T, et al. DEL-Dock: Molecular Docking-Enabled Modeling of DNA-Encoded Libraries[J]. Journal of Chemical Information and Modeling, 2023, 63(9): 2719-2727.

---

> ### Author Response · Authors · 2024-11-21
> **Thank you for your feedback!**
>
> Thank you for your positive comments and valuable feedback. We have addressed your comments in the responses below.
>
> **W1. “Why RF method performs the best on on-DNA set (Line 335)”, “why there are different performance rankings on on-DNA and off-DNA settings”, and “why DEL-Compose(M) and DEL-Compose(S) performs differently on on-DNA and off-DNA settings?”**
>
> For discussion on the RF and DEL-Compose results, refer to our responses to Reviewer 6jAS (W4 and W5). Additionally, there are differences between on- and off-DNA data because off-DNA effects are unobserved in the DEL data (that is one of the tradeoffs for generating data at scale). However, it is good to see that some models do observe good prediction correlation to off-DNA data, though this will vary from target to target (and library to library). The difference between two variants of DEL-Compose will greatly vary between datasets because it depends on whether the important binding structures are localized to particular synthons or are larger molecular scaffolds. In the former case, we expect DEL-Compose$^{(S)}$ to work better, whereas in the latter case DEL-Compose$^{(M)}$ might be better because it explicitly represents the entire molecule.
>
> **W2. More views and new settings are required, such as case study of top-ranking candidates**
>
> Thank you for this suggestion. We believe that the core of this benchmark should be testing model performance under different data splits to examine models’ ability to generalize, with different levels of task difficulty. This is exactly what KinDEL provides with random, disynthon, and similarity-based splits evaluated on on- and off-DNA testing compounds (see the information about the new similarity-based split in the general response above). However, we like the idea of expanding benchmarks to new setups like the ones presented in the DEL-Dock paper. The subset ranking in KinDEL would not provide much meaningful information because the validation compounds are already close to the library distribution (see Figure 5b). Other evaluations in the DEL-Dock paper assumed the existence of a strongly binding motif being benzenesulfonamides, which do not have a clear correspondence to our targets. However, to enable creation of new benchmarks similar to those present in DEL-Dock, we publish an additional BCA dataset using the same compound library. For more details, refer to the general response above.
>
> **Q1. Not enough comprehensive view, more metrics/comparisons to DEL-Dock**
>
> DEL-Dock uses a very small library (100k) for training purposes which gives a much smaller training set size and the validation set is a set of curated public datapoints of unknown quality. BCA (the protein used in DEL-Dock) is known to have a chemical structure with known binding affinity (benzenesulfonamide). As discussed on pages 13 and 14 of DEL-Dock, there is a discrepancy between the presence of this common potency conferring motif in the training set and validation set for DEL-Dock that may substantially impact performance. This is likely the source of divergence in performance for methods between these papers. In KinDEL we have multiple distinct potency conferring scaffolds and thus suffer less from this problem.
>
> **Q2. Is it possible to provide an advanced version of KinDEL dataset with machine-aided of molecule docking poses?**
>
> Please refer to the general response. We are planning to release 3D poses for purposes of consistent model development. The first batch of the poses for the top molecules has been already uploaded. Thank you for the suggestion.
>
> ---
>
> We hope that we have addressed your concerns, and the inclusion of 3D poses will make our benchmark more valuable. Please, let us know if you have any further questions or concerns we can resolve to make your review even more positive. Thank you again for your time and valuable feedback.

---

> > ### Comment · Reviewer_63rq · 2024-11-21
> > **Thank you for your response**
> >
> > In general, I am fine with all the responses. For W1, Q1, and Q2, I believe they are reasonable and good.
> >
> > For W2, I look forward to new results from new settings.
> >
> > For Q2, I believe the pose information is much more valuable.
> >
> > The authors have solved my questions.

---

> ### Comment · Reviewer_63rq · 2024-11-24
> **Additional experiment about subset Spearman**
>
> Would you please provide a subset Spearman metric evaluation just as mentioned in [1]? It is because a benchmark paper should cover most existing metrics. Thank you.
>
> [1] Shmilovich, K., Chen, B., Karaletsos, T., & Sultan, M. M. (2023). DEL-Dock: Molecular Docking-Enabled Modeling of DNA-Encoded Libraries. Journal of Chemical Information and Modeling, 63(9), 2719-2727.

---

> > ### Author Response · Authors · 2024-11-27
> > **Response**
> >
> > Thank you for further suggestions. DEL-Dock [1] uses a different set of training and validation data. The subset in [1] is chosen specifically to address concerns about bias due to molecular weight shifts between the training data and the publicly acquired validation data. KinDEL does not suffer from these shifts as we provide validation sets that are sampled from within the library (“In-Library” set). We feel that the provided validation benchmarks are better aligned with the training data than in [1], and further filtering by molecular weight would reduce sample size for no significant benefit.
> >
> > Additionally, the new suggested experiments have finished, and the results are included in the new revised version of the paper. In this revision, we have added a new cluster data split that is based on molecular similarity for both targets, DDR1 and MAPK14. We have also included the results of Chemprop, which is considered a SOTA method for predicting molecular properties using molecular graphs as inputs. We believe that these new results improve the comprehensiveness of our benchmark.
> >
> > Furthermore, we have included binding poses for both DDR1 and MAPK14 in Appendix F. This visualization shows how molecules in the library are expected to bind to our targets. In particular, the DDR1 ligand forms extensive hinge interactions, which are very characteristic for kinase targets.
> >
> > Thank you again for your invaluable feedback. Please let us know if you have any further questions you want to discuss with us during the extended discussion period.

---

> > > ### Author Response · Authors · 2024-12-02
> > > **Thank you for the discussion**
> > >
> > > Thank you once again for your positive review and valuable feedback on our paper. As the discussion period is coming to an end, please let us know if you have any further questions or suggestions. We hope the updates meet your expectations and enhance the overall impact of the paper. If you find our clarifications and additional results satisfactory, we would greatly appreciate your consideration for a stronger recommendation for our work.

---

> > > ### Comment · Reviewer_63rq · 2024-12-02
> > >
> > > I am so sorry for my late reply. I totally understand your point.
> > >
> > > I have no other question, and I think this paper should be accepted if the additional experiments the authors claimed were added in the revised version.

---

### Official Review · Reviewer_6jAS · 2024-10-30

**Soundness:** 3
**Presentation:** 3
**Contribution:** 2
**Rating:** 5
**Confidence:** 3

**Summary:**

The paper presents KinDEL, one of the first large publicly available DNA-Encoded Library (DEL) datasets focused on kinase inhibitors, specifically targeting MAPK14 and DDR1 kinases. The papers benchmarks different machine learning methods for binding prediction.

**Strengths:**

1. the paper is clearly written and contains the experimental details in the appendix.

2. The dataset includes off-DNA data. The benchmark used different data split strategies.

**Weaknesses:**

1. This paper is a valuable contribution to DEL-based drug discovery. It may serve as a good resource for computational drug discovery. But I am not sure about the importance of this paper for the ICLR community.

2. The dataset has only two targets and both are kinase. The biophysical assay validation set is relatively small.
3. The authors did not mention the library size or sequence depth of the DEL dataset. Does it have an effect on the dataset?
4. The authors show that DEL-Compose performs better for off-DNA data. It would be helpful to discuss the potential biases due to the DNA barcode.
5. Why does the RF method become worse for the disynthon split?

**Questions:**

See Weaknesses

---

> ### Author Response · Authors · 2024-11-21
> **Thank you for your feedback!**
>
> We appreciate your thoughtful feedback and have addressed each of your points below.
>
> **W1. I am not sure about the importance of this paper for the ICLR community.**
>
> We believe that DEL represents an emerging data modality with significant potential as a resource for addressing complex chemistry problems. In particular, one of the shortcomings of current foundational models trained on public data is that many models fail to generalize to particular targets. Because DEL data can generate a large amount of chemical data for any protein target, we can leverage this data modality to finetune these models. Another open problem in the field is how to correlate binding poses with actual binding affinity, and DEL data provides the supervision necessary to tackle this problem. To aid these efforts, we are updating the dataset to provide docking poses for the entire dataset. We are excited about the possibilities of this paired 3D pose and binding score dataset. This is why we believe that this data is a good contribution to ICLR and the machine learning community.
>
> **W2. The dataset has only two targets and both are kinase. The biophysical assay validation set is relatively small.**
>
> For the discussion about our targets, please refer to the general response above (we have added one non-kinase target). Regarding the size of the validation set, we believe that this number of validation compounds is reasonable for assessing model performance. To optimize the costs of compound resynthesis and biophysical assays, we selected a diverse set of molecules that cover the chemical space of the whole library (see the UMAP plot in Figure 5b). The validation set selection is discussed in Appendix C.
>
> **W3. The authors did not mention the library size or sequence depth of the DEL dataset. Does it have an effect on the dataset?**
>
> Library size is ~81M (which was described in section 2.2), the sequencing depth (number of read counts/sequence counts) is 514.5M/65.8M for MAPK14, 296.5M/49.8M for DDR1, and 440.7M/52M for BCA in all three replicates. We expect a difference between the two numbers as we amplify to install indices and the sequencing primers. Sequencing depth always has an effect on the data quality, as higher sequencing depth usually leads to better data quality (since there are more samples). However, this is often limited by resources, and we observe good correlation between replicates and counts (refer to Appendix B). We have added the information about sequencing depth to Appendix A.2.
>
> **W4. The authors show that DEL-Compose performs better for off-DNA data. It would be helpful to discuss the potential biases due to the DNA barcode.**
>
> We believe that DEL-Compose performs better for off-DNA data, because it contains the right inductive biases to regularize the model predictions. We know that individual data points in DEL experiments can be noisy, so it is important to not overfit onto individual molecules. DEL-Compose predicts a more conservative estimate of a molecule’s binding affinity, due to incorporating uncertainty directly in the predicted output distribution, which is why we believe DEL-Compose performs better on off-DNA data. We discuss briefly the differences between on- and off-DNA data in Section 4.
>
> **W5. Why does the RF method become worse for the disynthon split?**
>
> All models perform worse in the case of disynthon spits, and different models will observe different changes in these settings. Since specific structures are not observed as frequently in the disynthon splits, perhaps the random forest model is overfitting onto specific features, leading to a worse generalization.
>
> ---
>
> We hope that we have properly addressed your concerns. Please let us know if you have any further insights or questions answering which would make you feel more positive about our paper. Thank you again for your time and valuable feedback.

---

> > ### Comment · Reviewer_6jAS · 2024-11-26
> > **Thank you for your response**
> >
> > Thank you for addressing my concerns.
> >
> > I am still not very convinced about the contribution of this dataset. It is only specific to ICLR, not in general. Given the sequencing experiments, the generated dataset and the benchmark, this paper might be more suitable for a journal like NAR or Bioinformatics. About the *docking poses*, it seems that more issues need to be considered or made clear, which may beyond the scope of this paper. E.g. what docking tools will you use to generate the binding conformation? How can you determine the correct binding site? How to evaluate the quality of the docking poses?
> >
> > Regarding the model performance and the DNA barcode bias, will the bias of the on-DNA data affect the relative ranking of the binding affinity? How about training by the ranking loss instead of regression?
> >
> > **W5:** Sorry I did not describe it clearly. For the disynthon split, RF performs worse in the on-DNA data, but better in the off-DNA data than random split. Was there overfitting?
> >
> > **Question**
> > Does the count data reflect the on-DNA binding affinity? If yes, then the ranking should not be affected a lot, right?

---

> > > ### Author Response · Authors · 2024-11-27
> > > **Response**
> > >
> > > Thank you for continuing the discussion with us and providing more insights. Below, we respond to each of the points you raised.
> > >
> > > **1. Contribution to ICLR**
> > >
> > > The applicability of ML has grown so much in the last 10 years, especially in the domain of AI for science. This is an exciting new domain that provides a different data modality to traditional chemistry data, which is ideal for machine learning to make an impact. We submitted this paper under the “datasets and benchmarks” track, which is one of the topics explicitly listed in the call for papers for ICLR 2025. KinDEL is a new dataset in the growing field of the ML-aided analysis of DEL data, which is highly relevant in the modern landscape of drug discovery. Moreover, we propose a rigorous benchmark using this dataset, with multiple data splits and many models already tested in the benchmark. We also release the code that facilitates testing new methods. The focus on the evaluation of machine learning methods is what we believe makes this paper particularly interesting for the ICLR audience.
> > >
> > > **2. Quality of the docking poses**
> > >
> > > We agree that choosing correct binding sites, preprocessing both ligand and protein structures, and choosing the right docking algorithm are significant challenges when performing molecular docking experiments. That is why we provide ready-to-use docked poses so that ML scientists do not need to focus on solving these challenges. That also makes the benchmark model results independent of the chosen docking software and binding site. All the details on the docking experiments (including the used tools and protein structures) are already described in Appendix F. Example ligand poses are now also provided in Figure 8.
> > >
> > > The binding site selected for the docking experiments is the orthosteric binding site, which is the same part of the protein sequence as was screened in our DEL assay, for which we created protein constructs focused on these binding sites. It is still possible that some molecules might bind to other sites, but we expect this to be the vanishing minority of counts. Regarding the evaluation of the poses, we provide multiple poses for each ligand, which enables testing the ability of the models to select those poses that correlate best with the experimental binding data. As shown in the DEL-Dock paper, downstream models can be used to improve ranking of these hypothesized poses.
> > >
> > > **3. Training by the ranking loss**
> > >
> > > Yes, the bias of on-DNA data will affect the relative ranking of actual binding affinities. However, we can better understand this association by incorporating 3D information, as correct binding poses of on-DNA molecules have to have the DNA attachment point facing out of the pocket (since the DNA is too big to be inside the active site of the protein). These challenges actually make this data modality very suitable for machine learning, in order to understand these complex relationships (relating 3D geometries to count data). And again, DELs allow for many magnitudes of greater data generation compared to traditional chemistry screening data.
> > >
> > > Ranking loss is a great idea but has not been investigated in detail in the literature. We hope that datasets like ours can be used to develop these methods in the future.
> > >
> > > **4. Results of RF for on- and off-DNA data**
> > >
> > > The RF model does perform worse on-DNA compared to off-DNA in a couple of instances (disynthon split for MAPK14), which can be due to overfitting. It could also be that the model wasn’t optimized well enough for any one particular experimental setting (the model results are reported over many splits and experiments). Finally, the disynthon split is the most challenging among our data splits because some combinations of synthons are only present in the test set. It is possible that the on-DNA molecules selected for the held-out set are more difficult to predict in this setup for MAPK14 (containing more distinct disynthons) than the selected off-DNA molecules. To emphasize the differences between on- and off-DNA data, we have rewritten the “Held-out Test Set” description in Section 3.1.
> > >
> > > **5. Does the count data reflect the on-DNA binding affinity?**
> > >
> > > The count data is a reflection of on-DNA binding affinity, though there are some caveats. For instance, PCR bias might affect the true ranking of the binding affinities, by uplifting the counts for molecules based on their DNA tags (which is a confounding variable). We have added a short discussion about the noise sources in the new “Challenges and Future Directions” section on page 9.

---

> > > > ### Author Response · Authors · 2024-12-02
> > > > **Thank you for the discussion**
> > > >
> > > > Thank you again for your thoughtful feedback on our paper. We hope we have addressed all your concerns. We’ve worked hard to create a dataset that meets the highest standards and is easily usable by the ML community. If you have any further questions, we’d be happy to address them. As the discussion period is closing soon, we kindly ask if you could re-evaluate our paper in light of the updates, which include a new cluster-based split and the results of the Chemprop model (Tables 1 and 2), as well as the provided docking poses discussed in the new Appendix F.

---

### Official Review · Reviewer_BHqP · 2024-11-04

**Soundness:** 3
**Presentation:** 3
**Contribution:** 2
**Rating:** 5
**Confidence:** 3

**Summary:**

This study introduces a dataset of DNA-Encoded Libraries (DEL) focused on two specific kinases: Mitogen-Activated Protein Kinase 14 (MAPK14) and Discoidin Domain Receptor Tyrosine Kinase 1 (DDR1). Although the DEL datasets have proven valuable in drug discovery, they are relatively scarce for public use. The introduced dataset, named KinDEL (Kinase Inhibitor DNA-Encoded Library), comprises 81 million small molecules tested against MAPK14 and DDR1 kinases. An experimental evaluation is provided, comparing the performance of the proposed method in both on-DNA and off-DNA scenarios.

**Strengths:**

+ The availability of a well-curated and publicly accessible dataset is a notable contribution on its own, making this work a valuable resource for the research community.

+ The authors have conducted a thorough review of the relevant literature, effectively establishing the originality and motivation of the study, and demonstrating a clear understanding of the current state of the field.

**Weaknesses:**

- Given that the primary contribution of this work is a dataset, it would be beneficial to evaluate state-of-the-art (SOTA) methods on it, both to assess their performance in a new context and to demonstrate the dataset's comprehensiveness. The methods tested seem mostly old ones.

**Questions:**

- Referring to Section 5.1 Current Datasets, there have been, especially recent, efforts of providing DEL datasets even though they don't exactly match the features offered by the present study. However, it might possible that these existing datasets could be adapted to resemble KinDEL.Could you elaborate on whether KinDEL is novel in that sense, i.e., for instance, enhancing the dataset from [Iqbal et al. (2024)] by incorporating on-DNA synthesis can be challenging?

Sumaiya Iqbal, Wei Jiang, Eric Hansen, Tonia Aristotelous, Shuang Liu, Andrew Reidenbach, Cerise Raffier, Alison Leed, Chengkuan Chen, Lawrence Chung, et al. DEL+ ML paradigm for actionable hit discovery–a cross DEL and cross ML model assessment. ChemRxiv
doi:10.26434/chemrxiv-2024-2xrx4, 2024.

- Furthermore, how does KinDEL compare to the existing DEL datasets in terms of diversity, and how well does it reflect the performance of existing methods in predicting Poisson enrichment? Does KinDEL offer a more comprehensive or representative testbed for evaluating these methods?"

---

> ### Author Response · Authors · 2024-11-21
> **Thank you for your feedback!**
>
> We sincerely appreciate the Reviewer's recognition of the value our benchmark brings to the research community. We are committed to presenting a comprehensive comparison of the machine learning models that are frequently employed for analyzing DEL libraries. In our study, we have incorporated a diverse array of methods, including fingerprint-based methods (e.g. RF, XGBoost, DNN), graph neural networks (GIN) and synthon-based models (DEL-Compose). We are in the process of adding Chemprop (a SOTA GNN) to the benchmark. We would greatly appreciate any suggestions you might have on additional models that would enhance our comparison!
>
> **Q1. It might be possible that these existing datasets could be adapted to resemble KinDEL.**
>
> The data from Iqbal et al. [4] is a useful resource for developing new models, but it is still missing a lot of information that we provide in KinDEL. Their data encompasses 3 different DEL datasets, DOS-DEL, HitGen, and MSigma. Two of the datasets, DOS-DEL and MSigma, are not uploaded yet (currently missing from their Github), and their HitGen data only has 700k examples out of the 1B member library, so unfortunately it is difficult to assess the properties of their data. Furthermore, only full SMILES strings are provided, and no decomposed synthon structures, which can be important for modeling (as we have demonstrated in our experiments). This dataset also does not include pre-selection information, which can also be useful when modeling. Their data does, however, include the inhibitor condition which is useful to distinguish between allosteric/orthosteric/cryptic binders, which can be valuable.
>
> Given the limited amount of available data, we do not see a way that this dataset could be used to supplement or replace KinDEL. One important distinguishing feature of our data is our inclusion of replicate data. This is very important due to the potential noisiness of the experimental data. We feel that including this in our dataset makes it a valuable testbed for future machine learning experiments.
>
> **Q2. How does KinDEL compare to the existing DEL datasets in terms of diversity, and how well does it reflect the performance of existing methods in predicting Poisson enrichment? Does KinDEL offer a more comprehensive or representative testbed for evaluating these methods?**
>
> Chemical diversity is difficult to characterize especially on libraries of this size, and is highly task-dependent. For instance, chemical diversity measured through Morgan Fingerprints can fail to distinguish property cliffs, which is a very challenging problem for small molecule tasks [6]. To make the computation of diversity more feasible for such large libraries, one can evaluate the diversity of each synthon group separately. However, as mentioned in the discussion, most DEL datasets do not release their synthon structures. Nevertheless, we would be happy to measure the similarity between their library synthons and ours if the data were made available. Moreover, some public datasets (e.g. [5]) provide binarized evaluation labels. While these datasets are great resources for the community, we feel that it is even more valuable to measure how well models rank compounds (Spearman on enrichment) than it is to report accuracy on a binary evaluation set. While Hou et al. and Gerry et al. both provide enrichment based methods, their libraries are smaller and thus lack the coverage and diversity of ours.
>
> ---
>
> We hope that our answers clarify the importance of the proposed benchmark. Do you have any further questions in the meantime while we are testing more models? Thank you again for your time and valuable feedback.
>
> **[5]** Iqbal, Sumaiya, et al. "DEL+ ML paradigm for actionable hit discovery–a cross DEL and cross ML model assessment." (2024).
>
> **[6]** Van Tilborg, Derek, Alisa Alenicheva, and Francesca Grisoni. "Exposing the limitations of molecular machine learning with activity cliffs." *Journal of chemical information and modeling* 62, no. 23 (2022): 5938-5951.

---

> ### Comment · Reviewer_BHqP · 2024-11-24
>
> I appreciate your efforts to address the points I raised.
>
> The utilization of state-of-the-art (SOTA) methods presents an opportunity to examine whether their established strengths and limitations are accurately represented. Actually, the study by Iqbal et al. (2024) also suffers from this issue by incorporating fundamental machine learning (ML) approaches, such as Random Forest (RF), Support Vector Machines (SVM), and Multilayer Perceptron (MLP), despite the significant advancements in the ML field. It is essential to acknowledge that these methods may still outperform others in the target problem, although this assumption warrants further investigation.
>
> Regarding my second question, diversity can be explored through two primary avenues: (1) explicit, probably chemical, features representing the data points, and (2) the performance of the tested ML algorithms. A crucial concern with existing datasets across various domains is the lack of assessment regarding dataset diversity. Specifically, when a dataset exhibits high similarity among its data points, an algorithm's performance may be attributed to its ability to cater to these similarities rather than demonstrating generalization. This oversight can lead to inaccurate claims of SOTA algorithms, highlighting the need for a more comprehensive evaluation of dataset diversity.

---

> > ### Author Response · Authors · 2024-11-27
> > **Response**
> >
> > Thank you for your feedback. We have uploaded a revised version of the paper, in which we have added the results of a new cluster data split based on chemical similarity. We have also included Chemprop, which is a graph neural network that is considered SOTA in multiple molecular property prediction tasks. Chemprop DMPNN uses bond message passing in a molecular graph that encodes both atom and bond features. We hope that these additional results will provide a more diverse view on the performance of various machine learning models and their ability to generalize.
> >
> > To provide more information on the diversity of our library, we have computed the diversity of each synthon position in our combinatorial library by calculating the average Tanimoto distance between all synthons at the given position in the library. The diversity is 0.73, 0.83, and 0.59 for synthons A, B, and C, respectively. These measurements are an indication that our synthons cover a diverse range of chemical space. For comparison, BELKA is another large DEL dataset published recently, and the diversities of their building blocks are: 0.57, 0.89, and 0.89 (overall similar coverage, one position less diverse and two positions more diverse in terms of this diversity metric). For more information on the library diversity, please refer to Figure 3, showing the distributions of selected molecular properties, and Figure 5b, showing the UMAP of KinDEL.
> >
> > Thank you for your insightful comments. We hope the new results address your concerns, and we're eager to discuss further with you if you have other comments or suggestions.

---

> > > ### Author Response · Authors · 2024-12-02
> > > **Thank you for the discussion**
> > >
> > > Thank you for your time and thoughtful comments on our paper. As mentioned, our dataset is significantly larger than most others and includes a diverse range of building blocks, covering a broad spectrum of chemical properties. We believe this diversity enhances its value for testing new methods. If you have any further questions or would like us to clarify anything, please let us know before the discussion period ends. We would also be grateful if you could re-evaluate the paper and consider increasing your score, provided we have satisfactorily addressed your comments.

---

### Official Review · Reviewer_Z3Ms · 2024-11-04

**Soundness:** 2
**Presentation:** 2
**Contribution:** 3
**Rating:** 6
**Confidence:** 3

**Summary:**

The authors have released a new dataset, KinDEL, based on DNA-encoded library (DEL) testing, specifically targeting two kinases, MAPK14 and DDR1. They conducted experiments on this dataset to test model performance.

**Strengths:**

The authors provide a substantial amount of new data.
The structure of the article is clear and easy to follow.

**Weaknesses:**

1. The evaluation is limited to only two kinase targets, MAPK14 and DDR1. Given that both targets are kinases, this dataset may have limited generalizability as a benchmark for models applied to broader, non-kinase targets.
2. The data-splitting method may ensure that compounds with the same disynthon do not end up in the same split, but it doesn’t fully prevent similar compounds from being grouped together, as disynthons do not necessarily represent the core structure of small molecules. Other approaches, such as scaffold-based or overall molecular similarity-based splits, may yield a more robust assessment.
3. Presentation Improvements: The table headers are somewhat confusing, making it unclear what the numbers in the table represent without reading the text. In Figure 3, "SP³" should be corrected to "sp³" for accuracy.

**Questions:**

1. Why use AI to reduce data noise? If DEL diverges significantly from reality, it may indicate instability or unsuitability for the current task. AI-based discriminative models are inherently inaccurate to some extent, so how effective is it to use one inaccurate method to adjust for another?
2. More explanation is needed for why certain chemical properties in Figure 3 are considered "drug-like". For instance, the molecular weight peak exceeds the traditional threshold of 500, and the QED values are relatively low.
3. Why is on-DNA data significant here, when off-DNA structures are more relevant for practical applications like drug development? On-DNA structures are unlikely to be developed as drugs.
4. Why didn’t the authors test more end-to-end models? Also, why did they use Morgan fingerprints as input instead of molecular representations like SMILES strings (1D), molecular graphs (2D), or atomic coordinates (3D)?

---

> ### Author Response · Authors · 2024-11-21
> **Thank you for your feedback! (1/2)**
>
> We appreciate your detailed feedback and have addressed each point to clarify and improve our manuscript.
>
> **W1. The evaluation is limited to only two kinase targets.**
>
> We have added an additional target for the purposes of testing generalization. Please, refer to the general response above.
>
> **W2. The data-splitting method may ensure that compounds with the same disynthon do not end up in the same split, but it doesn’t fully prevent similar compounds from being grouped together.**
>
> We agree that segregating similar compounds into a single split is extremely important. In our experience, traditional scaffold methods (Bemis-Murcko) have substantial weaknesses in DEL data due to the combinatorial nature of the chemistry. To illustrate this we have calculated the number of unique scaffolds and their frequency in the top 1M for mapk14. There are ~300k unique scaffolds, and the most frequent one occurs 10,061 times and is a benzene. Please see the new appendix section “Dataset splitting” where we attach a plot of the frequency of the top 100 most common scaffolds (showing a rapid decline in frequency). We also show the six most common scaffolds, all of which are generic ring structures.
>
> Internally we have created a similarity based splitting method. It first uses UMAP to reduce 1024 ECFPs to 10 dimensions and then uses HDBSCAN to cluster them before constructing splits out of the scaffolds. This method is elaborated on in the new Appendix E. We will include an additional split from this method in our updated dataset. We are currently training models on this new data split.
>
> **W3. Presentation Improvements.**
>
> We apologize for any confusion caused by the table headers and appreciate your pointing out the typographical error in Figure 3. We have revised the table headers to improve clarity by adding the metric name “Spearman’s $\rho$”. We have also corrected "SP³" to "sp³."
>
> ---
>
> **Q1. Why use AI to reduce data noise?**
>
> DEL data efficiently generates hundreds of millions of data points, which is traditionally not possible through other screening methods because it would be prohibitively expensive. Despite its bias, DEL data routinely uncovers high affinity binders [3], which is why the data is a very valuable resource. Thus, even models that simply replicate DEL data (with inherent noise) are useful. Given the combinatorial nature of the library, we hope that future methods will demonstrate further denoising. While we know individual data points might have noise, we are confident that aggregates of molecular clusters should reveal the correct signals. By leveraging information in these aggregations, we believe AI models can recover information about the underlying affinities of the molecules tested.
>
> **Q2. More explanation is needed for why certain chemical properties in Figure 3 are considered "drug-like".**
>
> In Figure 3 we show the distributions of properties for KinDEL with ranges from Shultz (cited in the figure caption) as a reference. His paper updates the classic rule of 5 based on all drugs approved in the years since Lipinski et al. published their paper (1998-2017). In this paper, we see that the properties of “drug-like” molecules change over time. From 1998-2007, the 10th to 90th percentile range for molecular weight is 201 to 525, while from 2008-2017, this range shifted to 235 to 607. Generally, the trend in recent years has been towards larger drugs. Therefore, there is not necessarily a right or wrong range of molecules, and that’s why we think our data is still highly applicable for drug discovery.
>
> There is no discussion of QED within Schultz so we do not include ranges for it. While QED is a popular metric, it was derived from pre-2012 approved drugs and thus also is no longer entirely representative of current paradigms. Many of its constituent metrics are impacted by the current shifts in MW (HBA,HBD,PSA,ROTB,AROM). In this context, “low” scores mostly mean that DEL molecules do not tightly match the characteristics of 771 pre-2012 drugs.
>
> Generally, we intend Figure 3 as a reference to show that the distribution of properties of the molecules in KinDEL overlaps with those traditionally considered “druglike”. Of note is that Figure 2 of [4] also shows non-complete overlap with the “rules of 5” which suggests that this divergence is common in DELs.
>
> **[3]** Peterson, Alexander A., and David R. Liu. "Small-molecule discovery through DNA-encoded libraries." *Nature Reviews Drug Discovery* 22, no. 9 (2023): 699-722.
>
> **[4]** Gerry, Christopher J., et al. "DNA barcoding a complete matrix of stereoisomeric small molecules." *Journal of the American Chemical Society* 141.26 (2019): 10225-10235.

---

> > ### Author Response · Authors · 2024-11-21
> > **Thank you for your feedback! (2/2)**
> >
> > **Q3. Why is on-DNA data significant here, when off-DNA structures are more relevant for practical applications like drug development?**
> >
> > The observed count data in DEL experiments is an approximation of on-DNA $K_D$. By measuring on-DNA $K_D$ and validating our models against it, we are checking if the models can correctly predict the underlying properties of the molecules that confer the ability to bind as represented by the count data that models are trained on. This can be perceived as measuring how well models can remove the noise resulting from typical DEL problems like sequencing errors or competition between molecules in binding. As we intend this to be a benchmark, we hope future groups will eventually release models that outperform the Poisson baseline. This would be an indication of true denoising relative to the raw data.
> >
> > The prediction of off-DNA $K_D$ will be more challenging as the DEL data constraints the possible poses in the experiments by adding a DNA tag that needs to go to the solvent. However, if the models can find moieties in the molecules that confer binding while those molecules are tethered to the DNA, it is reasonable to assume that the majority of these moieties also would confer binding when these molecules are untethered from the DNA.
> >
> > **Q4. Why didn’t the authors test more end-to-end models? Also, why did they use Morgan fingerprints as input instead of molecular representations like SMILES strings (1D), molecular graphs (2D), or atomic coordinates (3D)?**
> >
> > Thank you for this suggestion. All neural-network models in our benchmark are trained in an end-to-end manner, i.e. DNN and GNN are both trained with gradient descent between the featurization of molecules and the calculated Poisson enrichment, which is common for DEL models since all target counts (with replicates) and control counts need to be somehow combined. DEL-Compose is trained with gradient descent directly using target and control counts, but this is one of few models that can handle DEL data probabilistically. We are also in the process of adding Chemprop to the benchmark. Could you clarify what end-to-end means in this context, and if you consider these models end-to-end?
> >
> > In terms of molecular representations, the GNN (GIN) uses 2D graphs and Chemprop does too. There are many methods that could be tested, and we would encourage researchers to try string or 3D representations, which should be possible with the docked poses that we are providing now (see the general response above).
> >
> > ---
> >
> > We hope your concerns are resolved. In particular, we released another dataset for a new non-kinase target, and we are testing all models on a new similarity-based split. Do you have any further questions or concerns we can address in the meantime? Thank you again for your time and valuable feedback.

---

> ### Comment · Reviewer_Z3Ms · 2024-11-22
> **Response to authors' response**
>
> Thank you for addressing my concerns. While your responses have clarified several points, there are still issues I would like to discuss further.
>
> ### Regarding Q1: Noise Reduction in DEL Data
> I appreciate your explanation regarding the potential of future methods to address noise. However, I remain uncertain about this. What did the authors mean by "Given the combinatorial nature of the library, we hope that future methods will demonstrate further denoising."? What is the connection between "the combinatorial nature of the library" and the possibility of denoising? Could you elaborate on how these aspects are causally related?
>
> Moreover, could you outline the current widely accepted approaches—whether machine learning-based or manual/rule-based—that are used to alleviate noise in DEL data? From your response, it seems such methods may not yet exist or are not well-established. Is this an accurate interpretation? If so, how does this impact the current utility of DEL data for model training and downstream applications?
>
> ### Regarding Q2: The Term "Drug-like"
> I agree that limiting "drug-like" criteria to the Lipinski rule of five is outdated, and I commend your efforts to provide an updated context. However, I note that you have not identified specific or quantitative ranges to support the claim that KinDEL’s properties fall within established "drug-like" boundaries. Without such evidence, I find the use of the term "drug-like" to describe this dataset potentially misleading.
>
> ### Regarding Q3: On-DNA Data and Relevance to ML Research
> Thank you for your detailed response, which I found interesting. Your explanation highlights how KinDEL brings attention to challenges within the DEL and chemical biology communities, such as noise reduction, the combinatorial nature of DEL data, and the unique issues with on-DNA data. These points are particularly relevant for the machine learning community. If your intent is to engage ML researchers, I suggest discussing these challenges explicitly in the manuscript, especially in terms that are accessible to researchers from pure ML backgrounds. This could enhance the impact of your work by framing the dataset as not just a resource, but as an invitation to tackle unresolved problems in the DEL domain. Could you share your thoughts on this?
>
> ### Regarding Q4: End-to-End Models
> By "end-to-end models", I refer specifically to methods that use raw molecular representations (e.g., SMILES strings, 2D graphs, or 3D atomic coordinates) as inputs, rather than relying on precomputed molecular features like Morgan fingerprints. I understand now that you have included models with molecular graphs as inputs. Additionally, I am curious about the performance of widely used denoising methods—both ML-based and non-ML-based—on KinDEL. If applicable, could you provide comparative results or insights into how these methods perform relative to the approaches you tested?
>
> Thank you again for your detailed responses. I look forward to your insights on these remaining points.

---

> > ### Author Response · Authors · 2024-11-27
> > **Response (1/2)**
> >
> > Thank you for your careful review and further questions. We address each of them below.
> >
> > **Q1. What is the connection between "the combinatorial nature of the library" and the possibility of denoising? Could you elaborate on how these aspects are causally related? Could you outline the current widely accepted approaches that are used to alleviate noise in DEL data?**
> >
> > As mentioned earlier, each individual molecule might have some level of noise, but groups of molecules within a synthon should show more similar binding affinities. For instance, if molecules A, B, C all share one same synthon, but only molecule A has very high counts, perhaps we can attribute most of the counts to noise, if none of B or C have high counts (see also Figure 4 to understand how combinations of synthons tend to have similar enrichment, which is represented as lines in the 3D plot). This is why the combinatorial nature of the library can explicitly help with being able to denoise the data better. This is, of course, only one of the inductive biases we can incorporate into our models to better differentiate the signal from noise in the data. We can also use information in the matrix/control and pre-selection data to further help denoise the data.
> >
> > There are various approaches to denoising. The simplest is termed Poisson enrichment, which attempts to correct for non-specific binding of molecules (i.e. the matrix) [1]. This enrichment is computed as a ratio of the fitted Poisson distributions over the control and target data. A more nuanced approach is that of deldenoiser, which attempts to learn corrections based on the data, modeling the binding process as a rate equation [2]. However, these approaches essentially compute summary statistics of the data, and are not based on the compound structure, so there are no generalization capabilities to new structures. DEL-Compose constructs explicit representations of mono-, di- and  tri-synthons and learns the true enrichment as a latent variable which is used to model the observed data [3]. In DEL-Compose, the learned enrichment is viewed as a denoised binding affinity of the molecule. In general, there is no one “correct” way to analyze DEL data, and we think this space still has much room for growth and we hope that releasing the dataset will enable scientists without wetlab access to participate in exploring it.
> >
> >
> > **[1]** Christopher J Gerry, Mathias J Wawer, Paul A Clemons, and Stuart L Schreiber. DNA barcoding a
> > complete matrix of stereoisomeric small molecules. *Journal of the American Chemical Society*,
> > 141(26):10225–10235, 2019.
> >
> > **[2]** Komar P, Kalinic M. Denoising DNA Encoded Library Screens with Sparse Learning. *ChemRxiv*. 2020; doi:10.26434/chemrxiv.11573427.v3 This content is a preprint and has not been peer-reviewed.
> >
> > **[3]** Benson Chen, Mohammad M Sultan, and Theofanis Karaletsos. Compositional deep probabilistic
> > models of DNA-encoded libraries. *Journal of Chemical Information and Modeling*, 64(4):1123–
> > 1133, 2024.
> >
> > **Q2. I find the use of the term "drug-like" to describe this dataset potentially misleading.**
> >
> >
> > While it's challenging to precisely define "druglike," over 30% of our library aligns with Schultz's established ranges (see Figure 3), providing numerous candidates for further optimization. Designing diverse combinatorial libraries naturally results in some compounds not meeting all filter criteria. We have clarified in the paper that a significant portion of the library falls within the range of already approved drugs as defined by Schultz.
> >
> > In the last paragraph of Section 2, we have added: “Notably, over 30% of the molecules in our library fall within the property ranges of already approved drugs, as outlined by Schultz. While certain synthon combinations may result in compounds that fall outside these preferred ranges, DEL molecules primarily serve to provide initial hits for drug discovery campaigns. These initial hits undergo iterative refinement during the hit-to-lead optimization process.” Thank you for your careful review and making this suggestion.

---

> ### Author Response · Authors · 2024-11-27
> **Response (2/2)**
>
> **Q3. These points are particularly relevant for the machine learning community. If your intent is to engage ML researchers, I suggest discussing these challenges explicitly in the manuscript, especially in terms that are accessible to researchers from pure ML backgrounds. This could enhance the impact of your work by framing the dataset as not just a resource, but as an invitation to tackle unresolved problems in the DEL domain. Could you share your thoughts on this?**
>
> We appreciate your suggestion, and we have explicitly added a paragraph in the discussion section labeled “Challenges and Future Directions” to address this:
>
> *“DEL data is powerful in that it specifically densely samples particular chemical spaces, which can be leveraged to learn more powerful representations. However, DEL data suffers from experimental noise to compensate for the scale of data this technology can generate. In particular, there are unobserved factors such as synthesis noise that makes it difficult to separate out signal from noise in the data (Zhu et al., 2021). Additionally, since our observations are sequencing read counts rather than actual binding affinity, the measurements also suffer from PCR bias (Aird et al., 2011). While we have presented several benchmark methods that try to learn a denoised enrichment from structure-based models, this is still an open question in the field, and we hope that our dataset release will enable the development of more denoising methods.”*
>
> Additionally, we have included more information about the differences between on- and off-DNA data in the description of the held-out sets in Section 3.1. The modified paragraph says:
>
>  *“The observed count data in DEL experiments are an approximation of the true on-DNA binding affinity ($K_D$). The count data are influenced by multiple sources of noise (see Section 4). We ultimately wish to rank molecules by binding affinity, so we use compounds with measured $K_D$ (from biophysical assays) as a test set. Performance on these compounds assesses if the models correctly rank compounds by $K_D$. This can be viewed as measuring how well models can remove the noise inherent to DELs.*
>
> *For both our targets, MAPK14 and DDR1, the selected compounds contained in the DEL library were resynthesized on- and off-DNA to create an in-library held-out test set. For hit finding, we would like to be able to predict off-DNA $K_D$. This is challenging because the DEL data comes from DNA bound molecules, and is biased by the DNA. The on-DNA $K_D$ more closely aligns with DEL data since the molecules in the training data are bound to the same DNA in the same way.  A few additional compounds were added from outside the library (and tagged with DNA) to create an additional held-out test set that we refer to as "Extended". The $K_D$ data from these biophysical assays are also released with our dataset. A UMAP visualization of the DEL including the in-library and external test set compounds is depicted in Figure 5b.”*
>
> **Q4. I understand now that you have included models with molecular graphs as inputs. Additionally, I am curious about the performance of widely used denoising methods—both ML-based and non-ML-based—on KinDEL. If applicable, could you provide comparative results or insights into how these methods perform relative to the approaches you tested?**
>
> Thank you for the clarification. One of the typical ways to analyze DEL data is with the Poisson enrichment developed by [1], which computes a ratio of Poisson distributions fit over the target and control data. We have included this in our benchmarks, under the row labeled “Poisson”. This is a computed metric of the data, and does not have any inference capabilities, so we use this number to understand the performance of trained structure-based models. What is exciting about these results is that some of our models, such as DEL-Compose, can make predictions with better performance than the Poisson enrichment, which is an indication that some models do have good denoising properties.
>
> We have also enhanced our benchmark by incorporating Chemprop, a model that utilizes molecular graphs as inputs, rather than traditional fingerprints (refer to Tables 1 and 2 in the updated revised paper). We hope that the inclusion of Chemprop addresses your concerns and improves the comprehensiveness of our benchmark.

---

> > ### Comment · Reviewer_Z3Ms · 2024-11-28
> > **Thanks for the authors' response**
> >
> > Thank you to the authors for their response. I have increased my score to 6. Large-scale, high-quality data mining is critical for the machine learning community and vice versa. I hope this work helps bridge the gap between the pharmaceutical, chemistry, and biology communities and the machine learning community.

---

> > > ### Author Response · Authors · 2024-12-02
> > > **Thank you for the discussion**
> > >
> > > Thank you for your thoughtful feedback and for increasing your score. Your insights were invaluable in improving the paper, and we deeply appreciate your contributions. If you have any additional questions or concerns that remain unanswered or could further enhance the paper, we would be more than happy to address them.

---

### Author Response · Authors · 2024-11-21
**(General Response) Thank you for your feedback!**

We would like to extend our gratitude for your invaluable feedback. Below, we have summarized the main concerns shared by the reviewers and how we want to address them:

**1. Only two kinase targets are included.**

We recognize the importance of increasing target diversity to make our benchmark more widely applicable, especially for methods tailored to different biological targets. Nonetheless, we firmly believe that our KinDEL dataset already serves as a significant resource for the development of new machine learning techniques focused on analyzing large combinatorial libraries. We have chosen two biological targets that not only complement existing public DEL datasets but are also well-studied and established within the scientific community (see for example the studies on the identification of MAP Kinase inhibitors confirmed with crystal structures [1] or on applying deep learning to generate new DDR1 inhibitors [2]). Notably, KinDEL comprises over 81 million compounds, significantly exceeding the size of typical activity datasets, and contains multiple targets opening new avenues for benchmarking, e.g., multitask learning methods. To further address the raised concerns, **we have expanded our dataset to include one more non-kinase target, Bovine Carbonic Anhydrase (BCA)**. This additional target not only enhances the diversity of our dataset but also offers new possibilities for testing model generalization. The experimental details and correlation plots between replicates of the BCA experiment have been added to the appendix, and the data is already available here: https://kin-del-2024.s3.us-west-2.amazonaws.com/data/bca_1M.parquet

**2. The benchmark could be improved by providing 3D data.**

Initially, we have not included any 3D data due to the computational cost of molecular docking experiments, which are only an approximation of molecule binding to rigid protein structures. However, after reconsidering the comments regarding the inclusion of 3D data, we decided that KinDEL could benefit from providing docked poses, which can be used to benchmark 3D models on a standardized set of poses. This way the results are not dependent on the docking procedure used by the authors of various methods. Therefore, to facilitate structure-based modeling, **we share 4.2M docked poses for the top 200k DDR1 and MAPK14 hits by target enrichment** from the recommended training sets:
https://kin-del-2024.s3.us-west-2.amazonaws.com/data/poses/2024-11-17_kindel-poses.sdf.gz

We plan to provide docked poses for the entire 81M library prior to publication. We have described our docking protocol in Appendix F of the revised paper. Thank you for this suggestion.

**3. More experimental results including models and new splits are recommended.**

We appreciate the Reviewers’ comments with ideas on how we can improve our benchmark. We computed a new similarity-based data split (see new Appendix E), and all the current models will be tested on this split before the discussion period ends. Additionally, we are training one more graph-based model, Chemprop, to showcase a better representation of state-of-the-art graph neural networks. We hope these additional experiments will significantly enhance the utility and value of our benchmark.

**[1]** Röhm, Sandra, et al. "Fast iterative synthetic approach toward identification of novel highly selective p38 MAP kinase inhibitors." *Journal of medicinal chemistry* 62.23 (2019): 10757-10782.

**[2]** Zhavoronkov, Alex, et al. "Deep learning enables rapid identification of potent DDR1 kinase inhibitors." *Nature biotechnology* 37.9 (2019): 1038-1040.

---

### Author Response · Authors · 2024-12-03
**Thank you for the discussion!**

Dear AC and Reviewers,

We want to thank you for the fruitful discussion during the discussion period. We are glad that all the concerns of Reviewers Z3Ms and 63rq were resolved, and they are recommending the acceptance of our paper. We have also made considerable efforts to respond to the feedback from Reviewers BHqP and 6jAS, though we have not yet received further comments from them following our recent responses and updates. The final revision includes the additional experiments we had run to address their reviews. To facilitate the further discussion between the Reviewers and the AC, we summarize the major changes in the revised paper.

**1. New experiments:** A new cluster-based data split and a more recent model in the benchmark.

We have run experiments on a new cluster-based split. Based on the results, this data splitting method is more difficult for the models than the random split but less difficult than the formerly proposed disynthon split. We have also added a new SOTA graph-based model, Chemprop, which is a more recent model that uses bond message passing. The new results are presented in the extended Tables 1 and 2. The new data split is also visualized in Figure 5a. The details on how the data split was computed and the discussion on the scaffold-based split are presented in Appendix E.

**2. Addition to existing dataset:** Docking poses provided along with the dataset.

High quality docking requires substantial expertise and specialized software in addition to significant computational resources. In order to democratize and standardize access to 3D binding poses for KinDEL, we have provided poses of the ligands docked to DDR1 and MAPK14. This addition makes it possible to evaluate structure-based predictors that use the 3D ligand poses using our benchmark. KinDEL is the first DEL dataset that provides docking poses, which makes this dataset useful for developing structure-based DEL models. The details on the docking procedure as well as examples of the docked compounds are presented in Appendix F.

**3. New target added to dataset:** A new non-kinase target (BCA) included in the dataset.

We have expanded our extensive dataset by introducing data for a new biological target, bovine carbonic anhydrase (BCA). This addition significantly enhances our collection, providing experimental counts in triplicate for all ~81 million compounds, which represents a 50% increase in experimental data. The inclusion of BCA enables researchers to test the generalizability of their models and explore new training methods, such as multi-task learning. Appendices A and B have been updated to detail the experimental procedures and confirm the replicability of BCA results. The data is available in the same S3 bucket as the existing data for the other two targets.

**4. More analysis of data and results:** A more detailed description of the role of the on- and off-DNA testing sets.

In response to the raised comments, we have added more discussion about the on- and off-DNA compounds in the “Held-out Test Set” section on page 4 and information of potential biases due to the DNA strand in the “Challenges and Future Directions” section on page 9. More discussion can be found in our responses to Reviewers Z3Ms and 6jAS.

---

In summary, KinDEL stands out as a comprehensive DEL dataset encompassing approximately 81 million compounds, evaluated against 3 targets, each with 3 replicates. This makes it significantly larger than other chemistry-related datasets, including other DEL datasets, with only Belka published concurrently being of slightly larger size. Our dataset provides full molecules as well as all three synthons used to build the library, offering unique opportunities for modeling applications. Moreover, KinDEL is the first DEL dataset to include docking poses, greatly enhancing its utility. We believe this dataset and its benchmark will serve as a valuable resource for the ICLR audience and the broader machine learning community.


Thank you again,

Authors

---

### Meta-Review · Area_Chair_WJPh · 2024-12-21

**Metareview:**

This work aims to present a large publicly accessible DNA-encoded library (DEL) datasets, referred to as KinDEL, which comprises two kinases: namely, Mitogen-Activated Protein Kinase 14 (MAPK14) and Discoidin Domain Receptor Tyrosine Kinase 1 (DDR1).
All reviewers recognize the value of such dataset, which may serve as valuable resources for advancing data-driven research in relevant areas.
The provided datasets and benchmarks are comprehensive and the construction process is also well thought-out and reasonable, building on the authors' comprehensive review of relevant literature.
However, the datasets are narrowly focused and not all reviewers are strongly convinced that the constructed datasets/benchmarks will be of interest to the broad ICLR community.
Performance assessment and analysis based on the latest SOTA methods are also insufficient and the paper also does not provide deeper insights regarding the observed evaluation results and the underlying factors of the respective models that led to the results.

**Additional Comments On Reviewer Discussion:**

The authors actively engaged with the reviewers during the discussion period to provide additional explanations and clarifications.
While the rebuttal has addressed part of the initial concerns raised by the reviewers, additional experimental results based on additional SOTA methods, further insights derived from the analysis results, and justification of the value of the presented datasets/benchmarks for the broader AI/ML community would be required for acceptance.

---

### Decision · Program_Chairs · 2025-01-22

Reject